# A natural biological adhesive from snail mucus for wound repair

Tuo Deng [1,2], Dongxiu Gao[1,3], Xuemei Song[1,2], Zhipeng Zhou[1], Lixiao Zhou[1,3], Maixian Tao[1,2], Zexiu Jiang[1,2], Lian Yang[1], Lan Luo[1], Ankun Zhou [1], Lin Hu[3], Hongbo Qin [1,3] & Mingyi Wu [1,2] ✉

The discovery of natural adhesion phenomena and mechanisms has advanced the development of a new generation of tissue adhesives in recent decades. In this study, we develop a natural biological adhesive from snail mucus gel, which consists a network of positively charged protein and polyanionic glycosaminoglycan. The malleable bulk adhesive matrix can adhere to wet tissue through multiple interactions. The biomaterial exhibits excellent haemostatic activity, biocompatibility and biodegradability, and it is effective in accelerating the healing of full-thickness skin wounds in both normal and diabetic male rats. Further mechanistic study shows it effectively promotes the polarization of macrophages towards the anti-inflammatory phenotype, alleviates inflammation in chronic wounds, and significantly improves epithelial regeneration and angiogenesis. Its abundant heparin-like glycosaminoglycan component is the main active ingredient. These findings provide theoretical and material insights into bio-inspired tissue adhesives and bioengineered scaffold designs.

Wound management remains a challenge in clinic because of the high incidence of traumatic injuries and refractory chronic wounds[1]. Surgical sutures and staples are gold standards for the reconnection of injured tissues and closure of wounds, which may cause pain, surgical site infection, and skin scarring[2]. Alternative treatments include tissue adhesives, such as fibrin glue and PEG-based adhesives, which are relatively effective and painless[3]. However, the currently available tissue adhesives still lack wet adhesion capacity and/or biocompatibility[4].

Over the last two decades, it is reported that some biologically inspired adhesives showed superior mechanical property and functionality[5,6]. The discovery inspired the design of active adhesives that may be promising candidates to replace sutures and staples. For instance, spherical nanoparticles from *English ivy* roots, composed of negatively charged glycoproteins, can form a penetrating film, enabling mechanical interlocking between the roots and the surface of substrates[7]. Inspired by the *English ivy* adhesion strategy, a super-strong aqueous synthetic tissue adhesive was fabricated using biodegradable polyurethane nanodispersions with opposite charges, which is proved effective for cerebrospinal fluid leak prevention and dura repair[8]. The development of catechol-based tissue adhesives was inspired by mussel-generated adhesive byssus fiber that is mechanically reinforced through protein-metal coordination via 3,4-dihydroxyphenylalanine (DOPA), it enables their anchorage in seashore habitats[9]. Moreover, the mussel-inspired adhesive material biopolymer-catechol can trigger the formation of barrier membrane for hemostasis by catechol-cation synergy, particularly, a catechol-conjugated chitosan can effectively promote hemostasis in animal bleeding models and human hepatectomy, independent of blood condition[10].

The mucus produced by the mollusc slug *Arion subfuscus* contains the intertwined anionic polymers and positively charged proteins[11]. The double-network hydrogel has strong toughness owing to the synergistic actions of two different polymer networks, which has great potential in guiding the design of next-generation tissue adhesives[12].

[1]State Key Laboratory of Phytochemistry and Plant Resources in West China, Kunming Institute of Botany, Chinese Academy of Sciences, 650201 Kunming, China. [2]University of Chinese Academy of Sciences, 100049 Beijing, China. [3]Key Laboratory of Chemistry in Ethnic Medicinal Resources, State Ethnic Affairs Commission & Ministry of Education of China, Yunnan Minzu University, 650031 Kunming, China. ✉e-mail: wumingyi@mail.kib.ac.cn

For example, it inspired the design of a tough two-layer adhesive consisting of an adhesive surface and a dissipative matrix, which has adhesion energy of more than 1000 J m$^{-2}$, far stronger than most commercial tissue adhesives[13]. In short, bionic scientists are advancing medical technology and tissue engineering by studying the internal mechanisms of adhesion in nature.

The mucus secreted by snails allows them to maintain conformal contact with the smooth surfaces of rocks and trees when they crawl[14]. More than 2000 years ago, land snails and their mucus were used to treat pain related to burns, abscesses, and other wounds[15,16]. Based on its natural adhesion behavior and indistinct bioactivity, snail mucus may be a potential natural biological adhesive for wound repair. Herein, we identified a natural biological adhesive from snail secretion, analyzed its composition and structure, elucidated the mechanism of its gelatinization, evaluated in vitro adhesion properties and in vivo effects on wound healing.

Normal wound repair is a highly organized biological process after treatment with sutures or tissue adhesives[17]. However, chronic wounds, such as diabetic foot ulcers (DFUs) and pressure sores, do not respond well to the current treatment because of their complex pathogenesis, continuous inflammation stage, and dysfunctional epithelialization[17]. We investigated the effects of snail secretion on the healing of chronic wounds in a diabetic rat model, compared with Alginate a commercial dressing. The mechanism of addressing inflammation was further explored using histological staining, immunofluorescence staining, and transcriptomic analysis.

In this work, we reveal the function and mechanism of natural bioadhesives from snail mucus in wound repair. These findings have important implications for understanding the adhesion mechanism of snail-mucus hydrogels and fabricating next-generation bio-inspired adhesives.

## Results

### Characterizations of snail mucus gel

As terrestrial mollusks, snails secrete mucus when crawling or irritated. In this study, snail-mucus gel (SMG) was collected from two species of snails (Fig. 1a, e). The fresh SMG from *Achatina fulica* appeared as a translucent hydrogel (Fig. 1b) and that from *Helix lucorum* was a milky-white hydrogel (Fig. 1f), both possessing deformability and extensibility. After collection, the SMG was immediately freeze-dried into spongy solid, named dried snail-mucus glue (d-SMG) (Fig. 1c and g). The average d-SMG yields from *A. fulica* and *H. lucorum* SMG were 3.65% and 5.26%, respectively. We found that the d-SMGs could form a glue with strong adhesive strength between the wet fingers rapidly within 15 s (Fig. 1d, h).

Subsequently, the elasticity of the snail-mucus hydrogel was studied using rheological tests. The storage modulus (G′) was higher than the loss modulus (G″), indicating the characteristics of an elastic network (Fig. 1i–j). Furthermore, the G′ of d-SMG was substantially increased (from 0.15 kPa to 17.8 kPa) with the decrease of hydration from 95% to 60%, which endowed the d-SMG with a wider applicability.

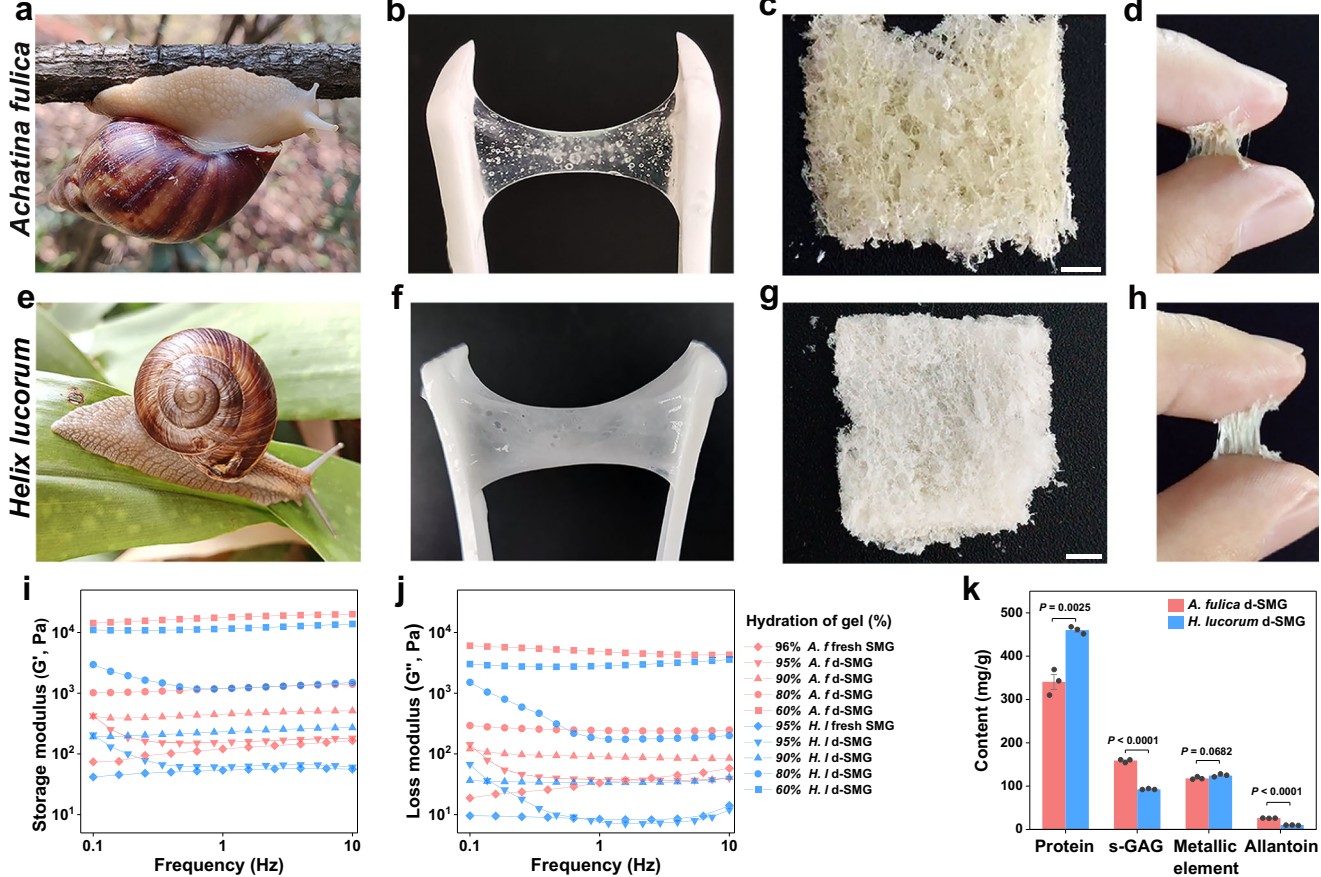

**Fig. 1 | Characterizations of d-SMGs. a–d** Photograph of *A. fulica* (**a**), fresh mucus from *A. fulica* (**b**), d-SMG from *A. fulica* (**c**), and adhesion of *A. fulica* d-SMG on skin (**d**). **e–h** Photograph of *H. lucorum* (**e**), fresh mucus from *H. lucorum* (**f**), d-SMG from *H. lucorum* (**g**), and adhesion of *H. lucorum* d-SMG on skin (**h**). **i–j** Storage modulus (**i**) and loss modulus (**j**) of SMG and d-SMG with different hydration ratio (red for *A.* *fulica*, and blue for *H. lucorum*). **k** The main components of the two d-SMGs (*n* = 3 biologically independent samples in each group). For **k**, two-tailed *t* test was used. Data are presented as mean values +/- SEM. Scale bars, 2 mm (images in **c, g**). Source data are provided as a Source data file.

To explore the mechanism of snail-mucus gelatinization, the chemical compositions of the two d-SMGs were analyzed. Both d-SMGs were rich in protein and anionic polysaccharides, as detected by specificity staining. Calcium salt was also observed in both d-SMGs as orange-red crystals after alizarin red S staining (Supplementary Fig. 1a). It is reported that DOPA-mediated protein-metal coordination is essential for the strength of byssal adhesion proteins from mussels and mollusks[9]. By contrast, d-SMG hydrolyzed samples did not contain L-DOPA, as detected by high-performance liquid chromatography (HPLC) and Arnow reaction staining (Supplementary Fig. 1b, c), indicating a different adhesion mechanism from that of mussels.

Chemical composition analysis of the anionic polysaccharide from d-SMG indicated that it's a sulfated glycosaminoglycan (snail glycosaminoglycan, s-GAG). Furthermore, quantitative analysis showed that d-SMG mainly contained protein (34.1% for *A. fulica*, 46.0% for *H. lucorum*), GAG (15.9% for *A. fulica*, 9.3% for *H. lucorum*), metallic element, and a small amount of allantoin (Fig. 1k and Supplementary Fig. 2) Surprisingly, d-SMGs had much higher content of s-GAG than the snail body (1.6%)[18]. The average weight (Mw) of s-GAG from *A. fulica* and *H. lucorum* was estimated to be 696.7 kDa and 540.7 kDa, respectively, and the sulfate ($-OSO_3^-$) content was 15.4% and 13.6%, respectively (Supplementary Table 1). Additionally, FTIR and NMR data (Supplementary Fig. 3) indicated that the s-GAG has a repeating sequence of →4)-α-GlcNAc (1 → 4)-α-IdoA2S (1 → , similar to the s-GAG from snail body reported previously[18,19]. In addition, the allantoin content of *A. fulica* and *H. lucorum* d-SMG was 2.65% and 1.03%, respectively (Supplementary Fig. 4), as determined by HPLC. Elemental analysis showed that the calcium contents of *A. fulica* and *H. lucorum* d-SMG were 2.64% and 6.67%, respectively (Supplementary Fig. 5).

The basic amino acid content of d-SMGs was close to that of the snail pleopod tissue, while the content of aromatic amino acids (phenylalanine and tyrosine) of both d-SMGs (-10%) were about two-fold of that of the tissue (-5%) (Supplementary Fig. 6 and Supplementary Table 2). More aromatic structural units can produce stronger intermolecular forces (π−π electron and cation−π interaction)[20]. Moreover, the d-SMG proteins were analyzed by liquid chromatography tandem mass spectrometry (LC-MS/MS). The main proteins in the *A. fulica* d-SMG were hemocyanin, achacin, actin, cytoplasmic, and mucin (Supplementary Tables 3–6), whereas those in *H. lucorum* d-SMG had similar but less diverse components. SDS-PAGE analysis of the d-SMG showed that the molecular weight of most proteins was above 100 kDa (Supplementary Fig. 6c). These protein bands (AF1–5 and HL1–6) were also analyzed by LC-MS/MS, and the results were consistent with the total protein of d-SMG (Supplementary Tables 7, 8).

Taken together, the d-SMG was mainly composed of proteins and polysaccharides of large molecular weights, metal ions ($Ca^{2+}$ and $Mg^{2+}$), and organic small molecular compounds (Fig. 1k and Supplementary Table 9). The basic and aromatic amino acids accounted for more than 25% in the two snail-mucus proteins, and the polysaccharide was a heparin-like sulfated GAG. Notably, compared with the source of invertebrate body and mammal tissue[18,19,21], the abundant heparin-like GAG in invertebrate mucus is rarely reported.

## Adhesion performance

Ideal bioadhesive materials improve biointegration and enhance tissue regeneration under pathophysiological conditions[22,23]. The high adhesion of hydrogel can prevent its detachment from the target tissue, thus promoting biointegration and healing[24]. The ability of fresh and dried SMG to adhere to wet tissues was evaluated. Fresh SMG (-200 mg) adhered immediately when contacted with wet fresh organs and could bear their weight (Fig. 2a). In addition, 3–5 mg of d-SMG was required to achieve the same effects, indicating the stronger adhesive force of d-SMG. Raman mapping showed that a diffusion region of about 0.3 μm was observed in the interface between d-SMG and the

tissue, owing to their remarkable differences of Raman scattering signals at 565 cm$^{-1}$(disulfide bond), 1090 cm$^{-1}$(aromatic ring and sulfonic acid), 1650 cm$^{-1}$(amide I), and 2930 cm$^{-1}$(C-H bond)[25] (Supplementary Fig. 7), indicating the mutual penetration between d-SMG and tissue. Therefore, interactions between d-SMG and tissues could include hydrogen bonds, π−π interactions, electrostatic interaction, and hydrophobic interactions, which contribute to the strong adhesion.

Given the dynamics of biological tissues and organs, both longitudinal shear stress and transverse tensile stress at the wound site can cause wound laceration[2]. Therefore, the bioadhesion properties of d-SMG (67% hydration) were evaluated by testing the shear strength, tensile strength, and interfacial toughness between two wet tissues according to American Society for Testing Material (ASTM) standards. Fibrin glues have been widely used as a hemostatic agent and tissue sealant in surgical applications since FDA approval in 1998[4]. However, fibrin glues also have some limitations, for instance, the risk of transmission of blood-borne disease, and the relatively poor mechanical properties with low cohesion strength[4]. Since d-SMG is also a bioadhesive, we selected fibrin glue as the main control in this study to evaluate the adhesion property of d-SMG. The shear strength of *A. fulica* and *H. lucorum* d-SMG was 23.64 ± 2.00 kPa and 35.7 ± 2.09 kPa, respectively, which is higher than that of fibrin glue (20.79 ± 0.51 kPa) (Fig. 2b–d). Additionally, when the adhesive time was extended to 1 h, the shear strength of d-SMG increased more than that of fibrin glue (Supplementary Fig. 8a). In addition, decrease of $Ca^{2+}$ content displayed no significant effect on the shear adhesion strength of d-SMG (Supplementary Fig. 8b and Table 10). In the tensile test (Fig. 2e–g), the adhesion strength reached 51.38 ± 3.18 kPa for *A. fulica* d-SMG and 82.59 ± 7.39 kPa for *H. lucorum* d-SMG, which were 90 and 145 times higher than that of fibrin glue (0.57 ± 0.09 kPa), respectively. Consistently, in the T-peel tests, their interface toughness (24.27 ± 1.37 J m$^{-2}$ for *A. fulica* and 33.7 ± 4.46 J m$^{-2}$ for *H. lucorum*) were 3.6 and 5 times higher than that of fibrin glue (6.77 ± 0.67 J m$^{-2}$), respectively (Fig. 2h–j). Moreover, the d-SMGs also proved effective in adhering blood-coated tissues. In summary, the d-SMGs exhibited better mechanical performance than clinical fibrin glue, indicating that they may be promising bioadhesives with potential clinical application.

## Biocompatibility and biodegradability

The optimal bioadhesive for wound repair should be biocompatible and biodegradable, therefore, these properties of d-SMGs were studied. In vitro cytotoxicity of the d-SMGs was evaluated by MTT method and real-time recording of cell counts. Compared with control, both d-SMGs had no obvious effect on the cell viability after treatment for 24, 48, and 72 h (Fig. 3a), and did not obviously affect the cell growth curve (Supplementary Fig. 9a). In the hemolysis assay, the hemolysis ratios of the two d-SMGs were all within the acceptable range (<5%) as biological materials, according to the Standard Practice for Assessment of Hemolytic Properties of Materials (ASTM F 756-17)[26] (Fig. 3b). Given that these snail-mucus gels contain heparin-like sulfated GAG, the activated partial thromboplastin time (APTT) of d-SMG and s-GAG were estimated. The results indicated that neither d-SMGs nor their GAGs exhibited APTT-prolonging activity (Fig. 3c), suggesting that they had no bleeding risk. These results suggest that the d-SMGs exhibit excellent cell and blood compatibility, and may be safe as tissue adhesives or wound dressings for further in vivo applications.

In vivo biodegradability and biocompatibility of the d-SMGs were assessed using a rat subcutaneous embedding model. The d-SMGs gradually disappeared within 21 d of embedding, indicating that d-SMGs can be fully degraded in vivo (Fig. 3d). Moreover, hematoxylin and eosin (H&E) staining showed that no obvious inflammatory responses were observed after d-SMG treatment. The skin incision in d-SMG-treated rats also healed completely without obvious scarring on day 21, further confirming the biocompatibility

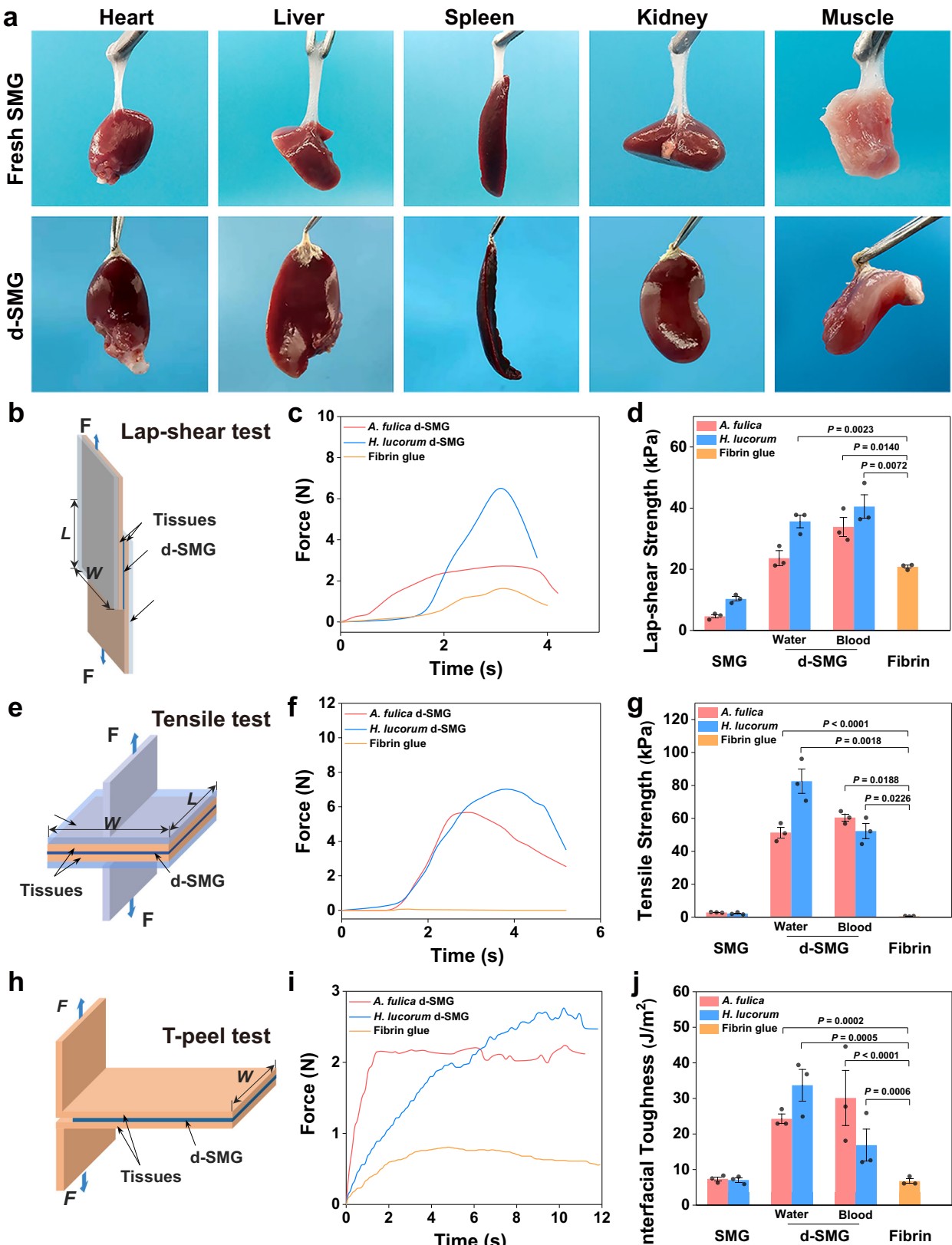

**Fig. 2 | Adhesive properties of d-SMGs. a** Wet adhesion properties of fresh SMG and d-SMG in rat fresh organs or tissues. **b** Schematic of the Lap-shear test. **c** Representative time-stress curves of the Lap-shear test. **d** Lap-shear strength of SMG and d-SMG ($n$ = 3 biologically independent samples in each group). **e** Schematic of the Tensile adhesion test. **f** Representative time-stress curves of the Tensile adhesion test. **g** Tensile strength of SMG and d-SMG ($n$ = 3 biologically independent samples in each group). **h** Schematic of the T-peel test. **i** Representative time-stress curves of the T-peel test. **j** Interfacial toughness of SMG and d-SMG ($n$ = 3 biologically independent samples in each group). For (**d**, **g** and **j**), two-tailed $t$ test was used. Data are presented as mean values ± SEM. Source data are provided as a Source data file.

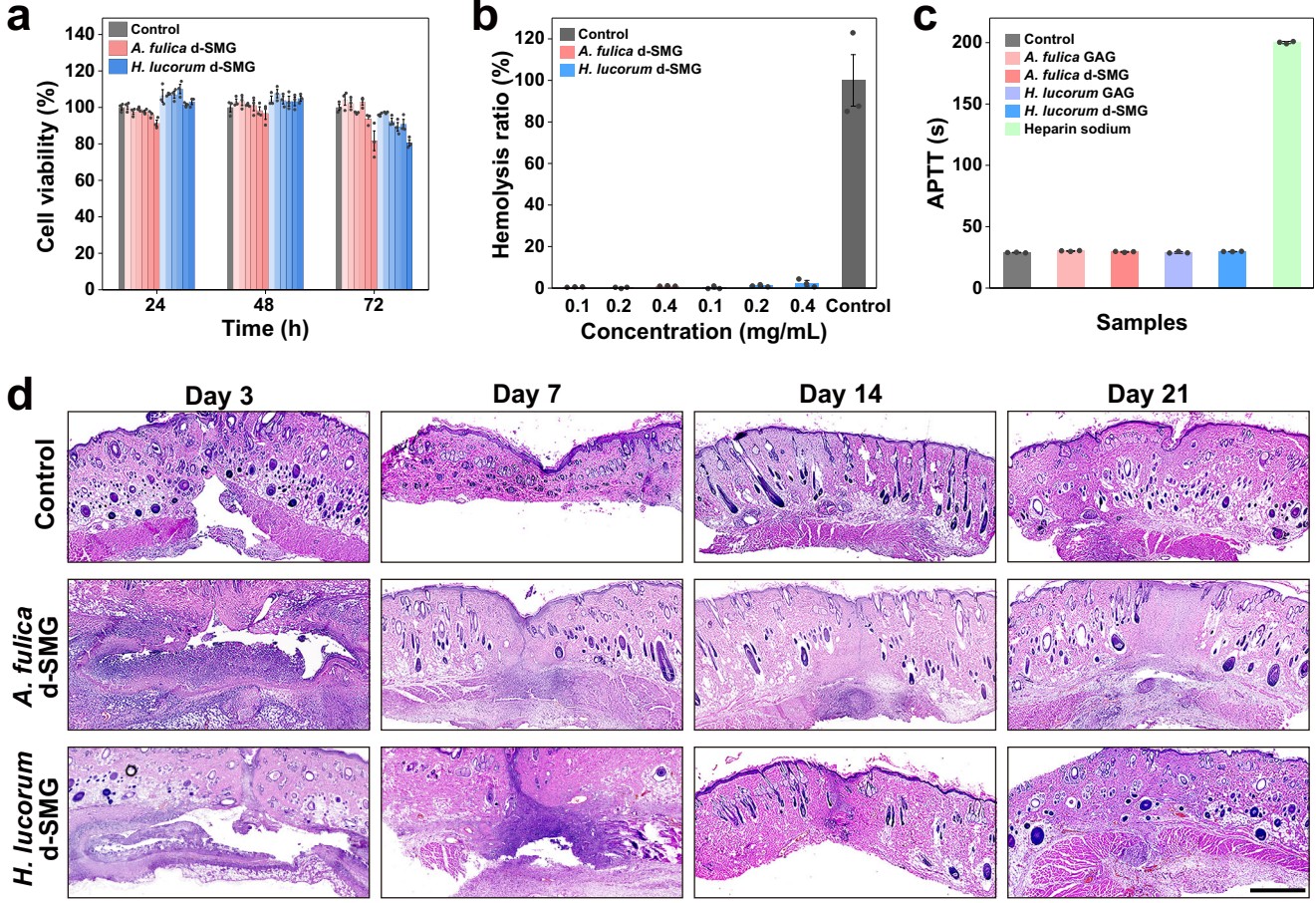

**Fig. 3 | Biocompatibility and biodegradability of d-SMGs. a** Effect of different concentrations (0.05, 0.1, 0.2, 0.5, 1.0, and 2.0 mg/mL from left to right in the graph) of d-SMG on cell viability by MTT method ($n = 3$ cells examined over 3 independent experiments in each group). **b** Effect of d-SMGs on hemolysis ($n = 3$ biologically independent samples in each group). **c** Effect of d-SMGs and their s-GAGs on APTT of human coagulation control plasma ($n = 3$ biologically independent samples in each group). **d** H&E staining images of skin tissue after subcutaneous embedding of d-SMGs in SD rat skin incision model ($n = 3$ biologically independent samples in each group). Data are presented as mean values ± SEM. Scale bar, 1 mm (images in **d**). Source data are provided as a Source data file.

(Supplementary Fig. 9b). These results suggest that in vivo biodegradability of the bioadhesive matched well with the skin wound healing process.

## Hemostatic properties in vitro and in vivo

Bleeding caused by injury, trauma, and surgical procedures may result in severe morbidity and mortality[27–29]. Uncontrolled bleeding is the main cause of death on battlefields and traumatic incidences[30]. Bioadhesive materials with hemostatic functions can help stop blood loss and promote wound closure[31,32]. Hence, the hemostatic ability of the d-SMGs was assessed using an in vitro dynamic blood-clotting index (BCI) test and in vivo rat liver bleeding model.

Notably, the d-SMG groups showed lower BCI than the gauze and negative control groups between 60 s and 150 s, which indicated that d-SMG accelerated blood clotting (Fig. 4a, b). We further investigated the effect of hemostatic materials on blood cells using scanning electron microscopy. Compared with gauze, more red blood cells and platelets adhered to the surfaces of d-SMG (Fig. 4c). These results demonstrate that the decreased BCI may be attributed to the strong adhesion of d-SMG to blood cells. In the rat liver bleeding model (Fig. 4d), the negative control group had the highest blood loss of 238.2 ± 24.90 mg (Fig. 4e, f). Treatment with d-SMG from *A. fulica* and *H. lucorum* dramatically decreased the blood loss to 113.0 ± 17.16 and 114.2 ± 9.60 mg, respectively. In contrast, gauze did not have a significant hemostatic effect (blood loss: 213.0 ± 46.06 mg). Apart from the hemostatic activity, d-SMGs could rapidly absorb liquid and form

an adhesive hydrogel in the bleeding site, which created a physical barrier that also accelerated hemostasis.

## Skin incision adhesion and wound healing in normal rats

In addition to hemostatic performance, tissue adhesion is vital for wound repair. The in vivo adhesive capacity of d-SMG was evaluated using a rat skin incision model, compared with a surgical suture and two commercial medical adhesives (Bioseal® containing porcine fibrin and COMPONT® containing cyanoacrylate (CA)). After immediate application, d-SMG showed obvious adhesive in the wet wound tissue, compared with the control (Fig. 5a, b). After *A. fulica* d-SMG treatment for 7 days the incised wound was tightly sealed, and *A. fulica* d-SMG showed better adhering effect than fibrin and CA. Although the wounds treated with sutures or fibrin glue healed completely, sutures or scabs were observed on the skin surface. As for CA treatment, the skin incision was well-bridged but the undegraded adhesive remained in the skin crevice might hinder tissue regeneration. The effects of suture and commercial medical adhesives were similar to those previously reported[20]. In addition, the d-SMG from *H. lucorum* showed poor performance in this in vivo adhesion test compared to that from *A. fulica*. (Supplementary Fig. 10). Consequently, *H. lucorum* d-SMG was not studied in the wound models.

H&E staining revealed that after d-SMG treatment for 7 d, the epidermis was completely regenerated, and there were evident hair follicles and sebaceous glands near the incision (Fig. 5c). Whereas, an apparent unrecovered dermis was observed in the wounds of the

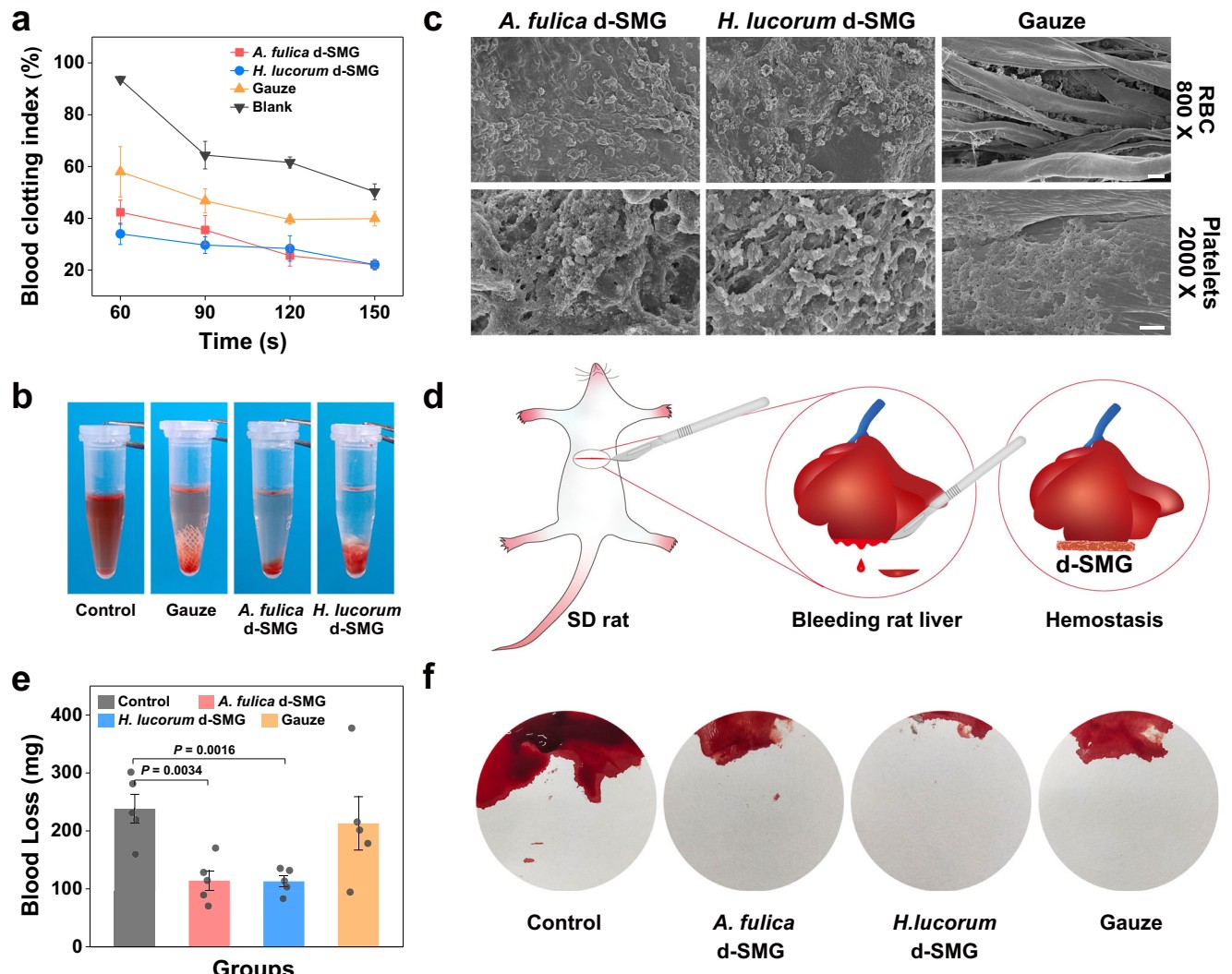

**Fig. 4 | Hemostatic properties of d-SMGs. a** Effect of d-SMGs on the dynamic blood-clotting index ($n = 3$ independent experiments in blank group; $n = 5$ biologically independent samples in *A. fulica* d-SMG, *H. lucorum* d-SMG and Gauze). **b** Photographs of blood clotting at 150 s. **c** Scanning electron micrographs of whole blood and platelets contacted with d-SMG (one biologically independent sample in each group). **d** Schematic for inducing bleeding on rat liver and stopping bleeding by applying d-SMG. **e** Effect of d-SMGs on blood loss in the rat liver bleeding model ($n = 5$ rats in each group). **f** Photographs of blood loss absorbed by filter paper in the rat liver bleeding model. For (**a** and **e**), two-tailed $t$ test was used. Data are presented as mean values ± SEM. Scale bars, 10 μm (images of RBC in **d**), 2 μm (images of platelets in **d**). Source data are provided as a Source data file.

control and fibrin groups on day 7. In the cyanoacrylate group, there were wide cavities in the wound, due to the undegraded remnant. The healing in the suture group was relatively better, except that the dermis was not completely healed. Masson trichrome staining showed that the collagen arrangement of *A. fulica* d-SMG-treated wounds was almost the same as that of normal skin, and epidermal regeneration was more complete than that of control, fibrin, and CA-treated groups on day 7 (Fig. 5d).

The effect of *A. fulica* d-SMG on wound healing was assessed using full-thickness skin wound model in healthy rats. The wound healing ratios of the *A. fulica* d-SMG and s-GAG group were 55.2 ± 2.09% and 56.4 ± 2.51% on day 5, respectively, which were significantly higher than that of the control group (45.9 ± 2.42%) (Supplementary Fig. 11a–d). While on day 11, s-GAG showed the better healing effect than other treatment, with the wound healing ratio of 92.7 ± 1.11%, significantly higher than that of the control group (83.2 ± 1.28%). The commercial alginate dressing (Alg™) displayed no obvious effect during the whole wound healing process. *A. fulica* d-SMG and s-GAG group had thicker granulation tissue on day 5 and more collagen deposition on day 15 as

analyzed by H&E staining and Masson's trichrome, respectively (Supplementary Fig. 11e–i). CD31 and alpha-smooth muscle actin (α-SMA) are biomarkers of vascular endothelial cell and smooth muscle cell, respectively. Immunofluorescence staining analysis of CD31 and α-SMA expression showed more neovascularization in the wound area of *A. fulica* d-SMG group (34.9 ± 3.3/mm²) and s-GAG group (26 ± 1.9/mm²) than that of control group (17.2 ± 1.1/mm²) on day 5 (Supplementary Fig. 12). These in vivo results indicated that both d-SMG and s-GAG effectively promoted wound closure and tissue regeneration.

**Chronic wound healing in diabetic rat models**

Chronic nonhealing wounds remain a major clinical challenge, and the development of pro-healing and cost-effective dressings is in urgent need[33]. Considering the excellent wound adhesion and healing performance of *A. fulica* d-SMG, its effects on diabetic wounds were studied by using a streptozotocin (STZ)-induced diabetic rat model[34] (Fig. 6a). Subsequently, the wound was treated with *A. fulica* d-SMG and its potential active ingredient s-GAG, compared with Alg™.

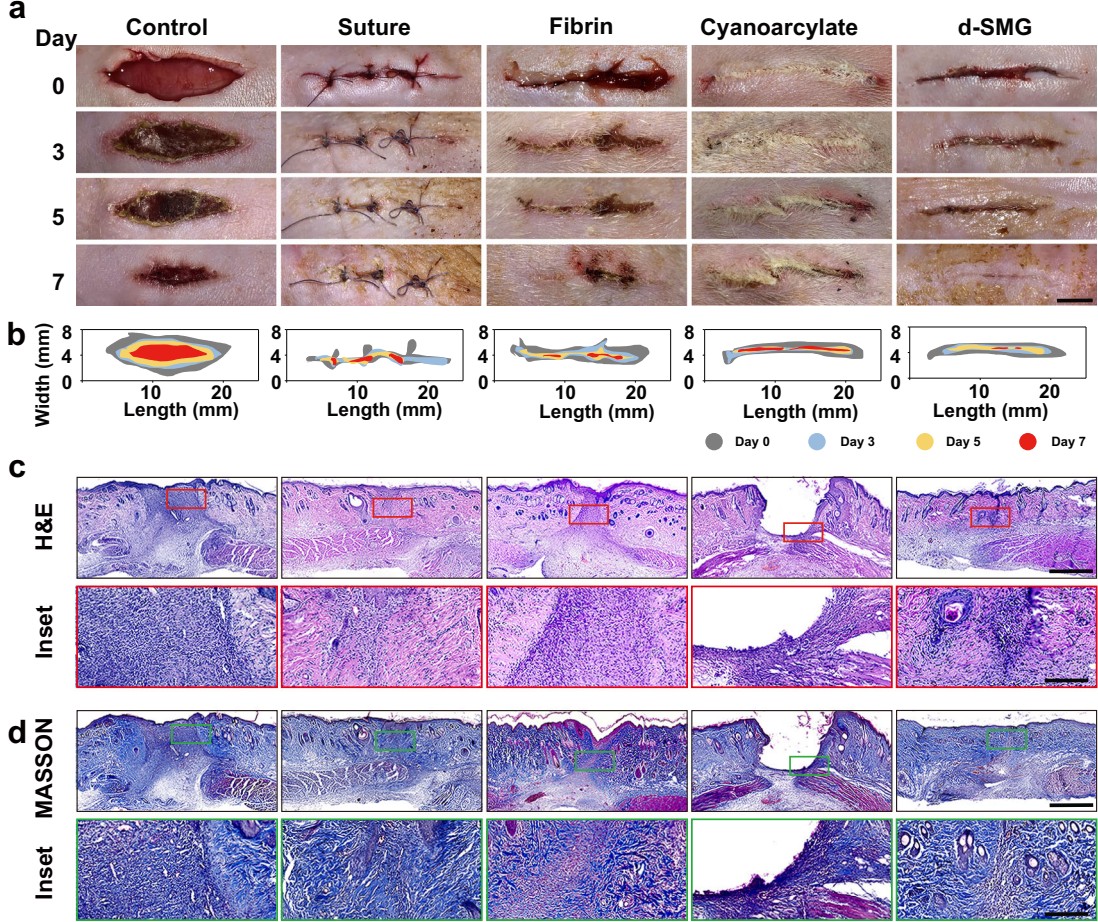

**Fig. 5 | In vivo wound adhesion efficacy of *A. fulica* d-SMG. a** Photographs of the rat incised skin after treatments. **b** Schematic of the dynamic wound healing process on day 0, 3, 5, and 7. **c, d** H&E staining (**c**) and Masson staining (**d**) of the wound sites on day 7 (*n* = 3 biologically independent samples in each group). Scale bars, 5 mm (images in **a**), 1 mm (images in **c, d**), 200 μm (inset in **c, d**).

Notably, the wounds treated with *A fulica* d-SMG (59.8 ± 2.47%) and s-GAG (63.3 ± 2.82%) showed significantly higher healing ratio than that of the control/saline (33.6 ± 2.69%) and Alg[TM] (46.5 ± 3.16%) groups on day 7 (Fig. 6b, c). The wound size in the d-SMG and s-GAG groups was gradually and substantially decreased within 14 days, suggesting that they effectively promoted chronic diabetic wound healing.

Neogenesis of granulation tissue is crucial for wound healing. Histological H&E staining revealed that *A. fulica* d-SMG and s-GAG significantly increased granulation tissue thickness, compared with the control (Fig. 6d, e). Furthermore, wounds treated with *A. fulica* d-SMG and s-GAG generated more epithelial tissue and hair follicles with adjoining sebaceous glands within the wound bed on day 14 (Fig. 6f). *A. fulica* d-SMG and s-GAG treated group exhibited significantly higher collagen deposition than that in the control group on day 7 and 14 (Fig. 6g). Vascular insufficiency is the major cause of diabetic ulcers[35]. To investigate the effect of *A. fulica* d-SMG on angiogenesis, immunofluorescence staining of CD31 and α-SMA was performed for histological analysis. Significantly increased number of vessels were observed in the *A. fulica* d-SMG and s-GAG groups on day 7, compared with those in the control and Alg[TM] groups (Fig. 6h–j). The results showed that *A. fulica* d-SMG and s-GAG could promote neovascularization and accelerate wound healing. The number of wound vessels were decreased on day 14 compared with that of day 7, which may indicate the transition from proliferative phase to remodeling phase, according to the angiogenesis pattern during wound healing[36,37].

Overall, both the *A. fulica* d-SMG and s-GAG effectively promoted chronic wound healing in the diabetic rat model by accelerating granulation tissue regeneration, angiogenesis, and collagen deposition. It also suggests that s-GAG is the main active ingredient of *A. fulica* d-SMG.

## Inflammatory regulation in vivo and in vitro

Wound healing is physiologically well-orchestrated and typically progresses through the defined phases of hemostasis, inflammation, proliferation, and remodeling. However, this process is disturbed in diabetic wounds, marked by prolonged low-grade inflammation that slows or stalls wound healing[38]. Specifically, macrophages play a central role in regulating the inflammation and wound repair process, by differentiating into M1 (inflammatory) phenotype or M2 (anti-inflammatory) phenotype[39]. However, diabetes causes a dysfunctional macrophage response and impaired phenotype transition from M1 to M2[40]. In the diabetic wound model, both *A. fulica* d-SMG and s-GAG showed pro-healing effect, which may be owing to their contribution in promoting the transition of wound healing from the inflammatory stage to the proliferation stage. Herein, their effects on macrophage polarization were further investigated by analyzing expression of M2 and M1, maker CD206 (M2 marker) and CD86 (M1 marker)[17]. *A. fulica* d-SMG and s-GAG significantly increased M2 macrophages (CD206[+]) on day 3 and 7, compared with control (Fig. 7a, b and Supplementary Fig. 13a). In contrast, *A. fulica* d-SMG and s-GAG decreased the ratio of M1 macrophages (CD86[+]) on day 7 (Fig. 7c). These results indicate that

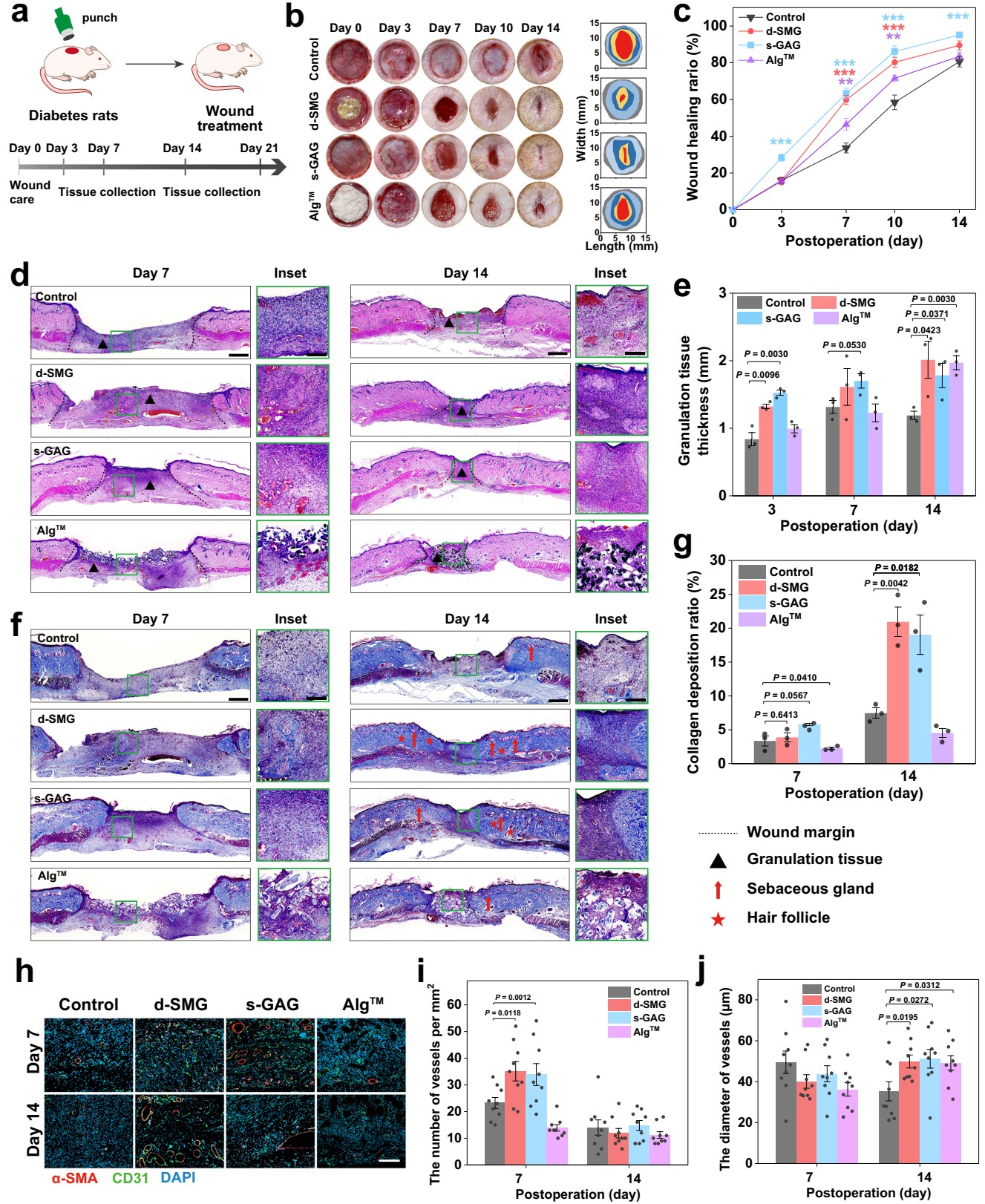

*A. fulica* d-SMG and s-GAG may potently stimulate polarization of macrophage to M2 type in chronic diabetic wounds, and further suggest that s-GAG is an important component d-SMG in inflammation regulation. The expression of interleukin-1β (IL-1β), a main inflammatory cytokine secreted by M1 macrophages, was significantly decreased in *A. fulica* d-SMG group compared with control on day 7

(Supplementary Fig. 13b, c). Moreover, *A. fulica* d-SMG reduced the levels of many other proinflammatory factors in diabetic wound, such as a 3-fold decrease in tumor necrosis factor-α (TNF-α) expression and a 5.7-fold reduction in interferon-γ (IFN-γ) and interleukin-6 (IL-6) expression (Fig. 7d and Supplementary Fig. 14). Similarly, inflammatory cytokines in the s-GAG group were also downregulated.

**Fig. 6 | Effects of *A. fulica* d-SMG and s-GAG on chronic wound healing in diabetic rats. a** Schematic illustration of the diabetic wound model in SD rats. **b** Representative images of the wound healing behavior and dynamic wound healing process on day 0, 3, 7, 10, and 14. **c** The wound healing ratio on day 0, 3, 7, 10, and 14 ($n = 7$ rats created over 2 independent wounds in each group, two-tailed $t$ test was used, *$P < 0.05$, **$P < 0.01$, ***$P < 0.001$). **d** H&E-stained images of the wound tissue on day 7 and 14. **e** The granulation tissue thickness in wound tissue on day 7 and 14 ($n = 3$ biologically independent samples in each group). **f** Masson staining of the wound tissue on day 7 and 14. **g** The collagen deposition ratio of wound tissue on day 7 and 14 ($n = 3$ biologically independent samples in each group). **h** Representative images of CD31 and α-SMA immunostaining in wound tissue on day 7 and 14. **I, j** Quantification of the number (**i**) and diameter (**j**) of vessels in the wound tissue ($n = 9$ biologically independent samples in each group). For (**c, e, g, i**, and **j**), two-tailed $t$ test was used. Data are presented as mean values ± SEM. Scale bars, 1 mm (images in **d, f**), 200 μm (inset in **d, f**), 100 μm (images in **h**). Source data are provided as a Source data file.

Additionally, the effects of *A. fulica* d-SMG and s-GAG on macrophage polarization were also investigated using RAW264.7 cells. Consistently, s-GAG significantly increased the macrophage M2 phenotype (CD206$^+$) more than 2 folds of control, slightly stronger than *A. fulica* d-SMG and weaker than the positive control IL-4, while Alg™ had no obvious effect (Fig. 7e, f). This indicates that *A. fulica* d-SMG and s-GAG indeed can regulate macrophage polarization to M2 type. Signal transducers and activators of transcription (STAT1, STAT3, and STAT6) are canonical transcription factors in regulating macrophage polarization and activity[41]. The activation of STAT1 promotes M1 macrophage polarization, resulting in inflammatory response, while activation of STAT3 and STAT6 contributes to M2 macrophage polarization, associated with immune suppression and tissue remodeling[41]. To further explore the mechanism, we analyzed the effects of *A. fulica* d-SMG and s-GAG on the signaling protein and mRNA levels in macrophages, by Western Blot and qRT-PCR, respectively. Compared with the control, *A. fulica* d-SMG and s-GAG increased the phosphorylation level of STAT-3, whereas they had no obvious effect on the phosphorylation of STAT1 and STAT6 (Supplementary Fig. 15). These results indicated that they may promote the polarization of macrophages to M2 by activating STAT-3. Moreover, they also significantly upregulated the mRNA expression levels of *Arg1* and *Il10*, which are associated with macrophage M2 phenotype (Supplementary Fig. 15).

Heparin-like compounds as negative-charged glycosaminoglycan can bind inflammatory cytokines and regulate the inflammatory microenvironment in chronic wounds[42,43]. Since the *A. fulica* d-SMG contains rich heparin-like GAG, capturing function of s-GAG may also result in reduction of inflammatory cytokines. The surface plasmon resonance (SPR) analysis showed that s-GAG strongly bond to inflammatory cytokines (TNF-α, IL-6, IL-8, and IP-10) with the affinity ($K_D$, 0.01–0.34 μM) similar to heparin ($K_D$, 0.17–0.61 μM) (Supplementary Fig. 16). Therefore, d-SMG reduced inflammatory cytokine levels in wound tissues by capturing function for their ability of binding cytokines, as well as by reducing secretion for their regulating macrophage polarization towards M2 anti-inflammatory phenotype.

To systematically assess the reversal of persistent chronic inflammation after treatment with *A. fulica* d-SMG, transcriptome of the wound tissue analysis was analyzed on day 3, 7, and 14, which covered the inflammatory and proliferative phases. After treatment with *A. fulica* d-SMG, 855, 236, and 728 differentially expressed genes (DEGs) were identified on day 3, 7, and 14, respectively, using edegR analysis (false discovery rate <0.05) (Fig. 8a, b and Supplementary Fig. 17). Specifically, the expression of genes related to inflammation, including *Tnf, Ccl5, Nos2, Il1b, Il18*, and *Tlr5*, were downregulated after treatment with *A. fulica* d-SMG on day 3 (Fig. 8c), which was consistent with the reduction in inflammatory cytokines (Fig. 7). In addition, the expression of genes related to wound healing and angiogenesis (*Vegfb, Tgfb1, Fgf3*, and *Col7a1*) was upregulated after *A. fulica* d-SMG treatment on day 7 and 14 (Fig. 8d). Gene ontology (GO) analysis showed 648 downregulated genes related to immune response, metabolic process, and oxidation-reduction process on day 3 (Fig. 8e). GO analysis of the 240 upregulated genes indicated that *A. fulica* d-SMG treatment enhanced organ regeneration, wound healing, and cell proliferation on day 14 (Fig. 8f). Subsequently, the relative mRNA

expression levels of the three groups were analyzed using real-time fluorescence quantitative PCR. In the *A. fulica* d-SMG group, the genes related to inflammation (*Cd86* and *Il-1b*) were significantly down-regulated ($P < 0.01$), whereas the genes pertaining to M2 phenotype macrophages (*Arg1, Il10, Cd163*, and *Jak3*), tissue regeneration, and angiogenesis (*Tgfb, Vegfa, Col1a1, Col3a1*, and *Pdgf*) were upregulated on day 14 (Fig. 8g).

These results demonstrate that *A. fulica* d-SMG regulates the inflammatory response in wound microenvironment by promoting the polarization of macrophages towards the anti-inflammatory M2 phenotype, and by capturing inflammatory cytokines during diabetic wound healing.

## Discussion

Each year, tens of millions of people suffer from various tissue wounds, ranging from minor skin cuts to severe injuries resulting from traumatic incidents, chronic wounds such as diabetic ulcers, and pressure sores[4]. Clinical treatments for these injuries involve reconnection and closure of injured tissues, mainly using sutures or staples. However, sutures and staples may cause additional tissue damage, scarring and/or secondary surgical infection. Biological adhesives are promising alternatives of sutures and staples. Based on the increasing knowledge of the natural adhesion phenomena, the development of biomimetic tissue adhesives has attracted increasing attention in recent years.

Mussel can bind to various substrates by secreting adhesive byssus fibers mechanically reinforced by a proteaceous adhesive containing DOPA[44]. The catechol group of DOPA can be oxidized to an active dopaquinone, leading to Schiff base reaction or Michael addition with free amino groups[45]. Catechol can also form rich ionic or covalent bonds with various substrates. These chemical reactions of the mussel adhesive produce strong cohesion and adhesion abilities. This discovery led to a surge in the development of catechol-based tissue adhesives. Further studies showed that mussel adhesion is an orchestrated process that includes protein fabrication, phase behavior, delivery, deposition, and assembly[46]. A recent study found that in stockpiled secretory vesicles catechol-based proteins were mixed with metal storage particles containing iron and vanadium ions, forming protein-metal bonds within the nascent byssus[9].

Unlike mussels, land snails do not permanently adhere to their habitat, and their movement depends on the coordination of their foot and adhesive secretions. Therefore, the absence of DOPA in snail mucus is not surprising (Supplementary Fig. 1); however, it is interesting that a gelatinous adhesive can still form without DOPA. Large molecular proteins and sulfated GAGs were the main components of the snail gel (Fig. 1). The protein in snail-mucus gel can create a three-dimensional skeleton, which interacts with linear s-GAG to form supramolecular entanglement, and its positively charged amino or guanidine groups can form electrostatic interactions with abundant negatively charged sulfate and carboxyl groups in s-GAG (Fig. 9a). In addition, hydrogen bonds, π-π interactions, and hydrophobic interactions are also widely present because of the high content of hydroxyl, aromatic, and aliphatic amino acids (Supplementary Table 1–2). Moreover, divalent cations (Ca$^{2+}$ and Mg$^{2+}$) in the mucus can adjust the elasticity of the gel through complexation and electrostatic interactions[12]. Consequently, the natural snail adhesive exhibits

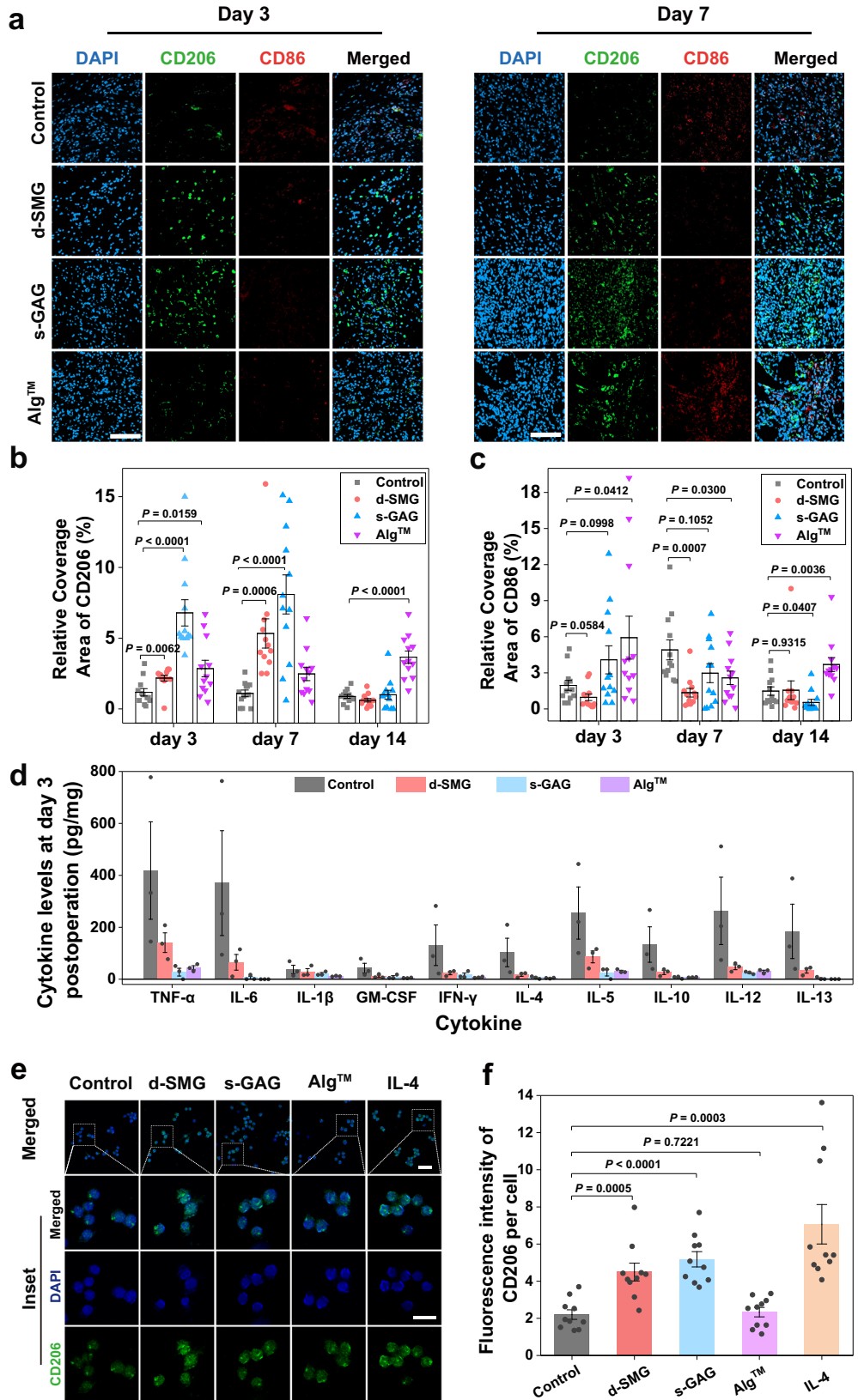

**Fig. 7 | Effects of *A. fulica* d-SMG and s-GAG for inflammatory regulation in vivo.**
**a** Representative image of CD206 and CD86 immunostaining in wound tissue on day 3 and 7. **b** Quantification of CD206+ area in the wound tissue (*n* = 12 biologically independent samples in each group). **c** Quantification of CD86+ area in the wound tissue ((*n* = 12 biologically independent samples in each group). **d** Cytokine levels in wound tissue on day 3 (*n* = 3 biologically independent samples in each group).

**e** Representative images of CD206 and DAPI staining of RAW264.7.
**f** Quantification of the fluorescence intensity of CD206 in RAW264.7 cells (*n* = 10 biologically independent cells in each group). For (**b**–**d** and **f**), two-tailed *t* test was used. Data are presented as mean values ± SEM. Scale bars, 100 μm (images in **a**), 50 μm (images in **e**), 25 μm (inset in **e**). Source data are provided as a Source data file.

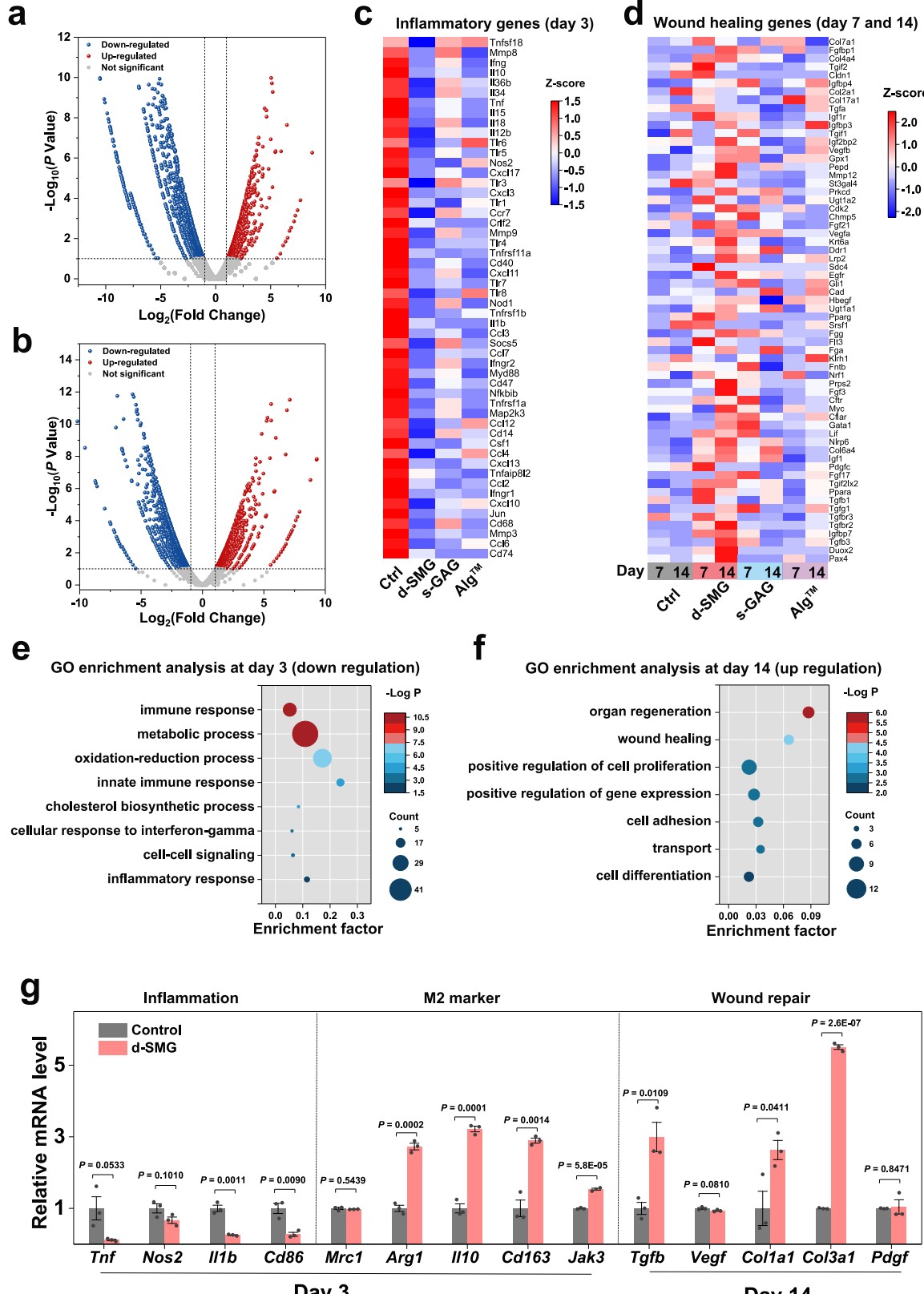

**Fig. 8 | Analysis of differentially expressed genes in diabetic wound tissue.**
**a, b** Volcano plots showing the upregulated and downregulated genes in d-SMG group on day 3 (**a**) and 14 (**b**). **c, d** Heat map of inflammation genes on day 3 (**c**) and wound healing genes on day 7 and 14 (**d**). **e** Gene ontology (GO) enrichment analysis of the downregulated genes on day 3. **f** GO enrichment analysis of the upregulated genes on day 14. **g** The relative mRNA level in the diabetic wound tissue evaluated using real-time qPCR (*n* = 3 biologically independent samples in each group). For (**a**, **b**), Wald test was used; For (**e**,**f**), Hypergeometric tests and Benjamini–Hochberg FDR control procedures were used; For (**g**), two-tailed *t* test was used. Data are presented as mean values ± SEM. Source data are provided as a Source data file.

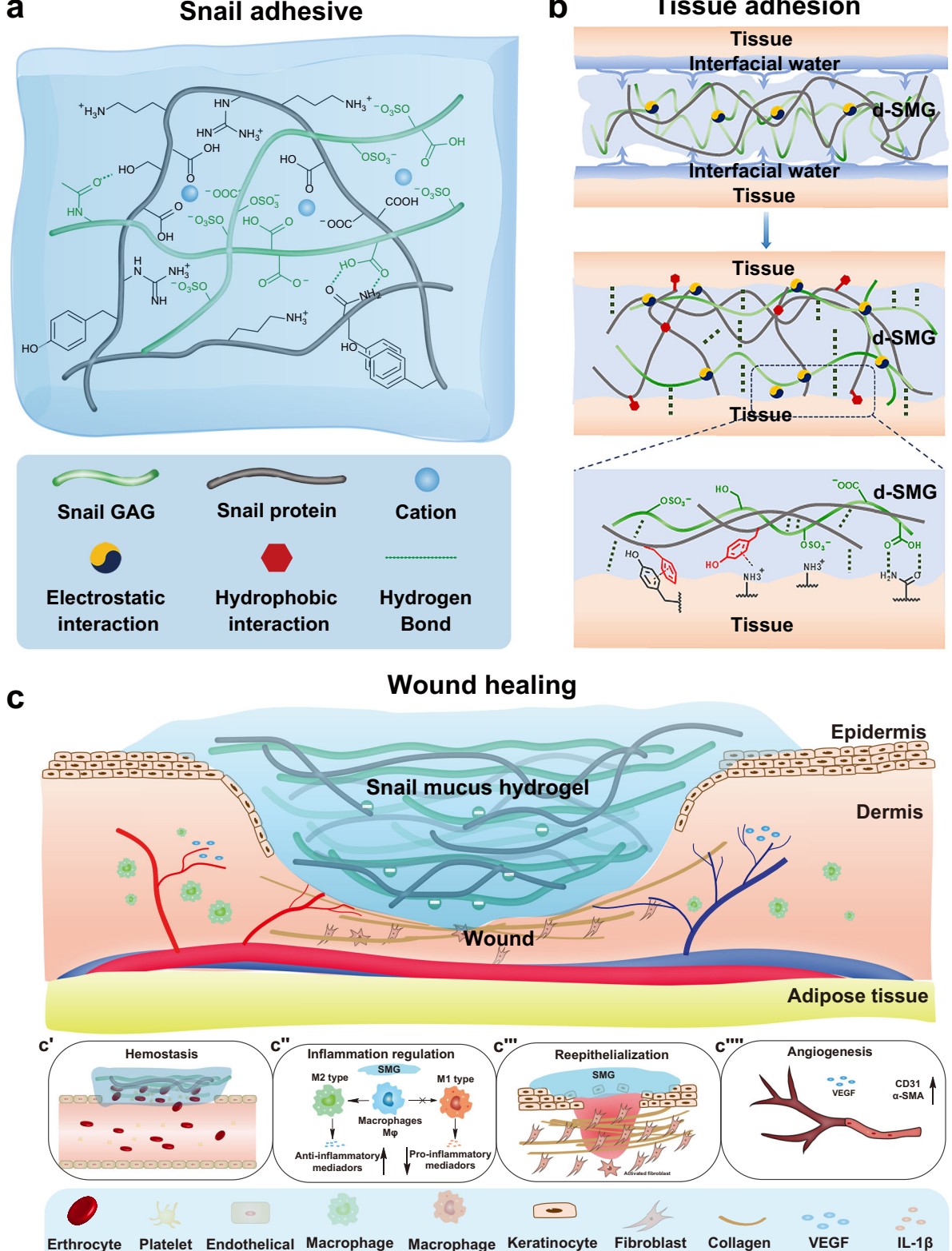

**Fig. 9 | Schematic interpretation of the mechanism d-SMG in wound healing.**
**a** Formation of double-network snail-mucus hydrogel. **b** A possible tissue adhesion mechanism of d-SMG. **c** The postulated mechanism of d-SMG in accelerating wound healing, including hemostasis (**c'**), anti-inflammation (**c''**), granulation and epithelium regeneration (**c'''**) and angiogenesis (**c''''**).

supramolecular synergy, and strong toughness and cohesion. Its gelation mechanism is similar to a double-network hydrogel produced by the slug *Arion subfuscus*[12], which has great potential in guiding the development of synthetic adhesives. For instance, a synthetic adhesive system, consisting of an adhesive surface and a dissipative matrix, had an interface toughness that was tens of times stronger than that of ordinary tissue adhesives[13,47]. Combining previous reports with our findings, the double-network hydrogel is likely a common phenomenon in gastropod secretions, which can inspire the development of tissue adhesives and wound dressings.

Adhesion to wet surfaces, including biological tissue, is important in numerous fields but extremely challenging. The adhesion between the two interfaces mainly depends on the formation of covalent bonds and intermolecular forces (hydrogen bonds, electrostatic interactions, and van der Waals interactions)[48]. However, on wet tissue surfaces, water seals the hydrogen bond receptors and donors of the two interfaces, preventing interactions. Sulfated GAG, a polymer in d-SMG, is hydrophilic because it is rich in sulfates, carboxyl, and hydroxyl groups, thus it can effectively absorb water, and by eliminating interface water the interactions between supramolecules (Fig. 9b) can be reestablished. Our data showed that d-SMG had stronger in vitro and in vivo adhesion to wet tissue than commercial fibrin glue (Figs. 2, 5). Fibrin glue and cyanoacrylate are two commonly used tissue adhesives in clinic, but they lack sufficient efficacy for chronic wound healing such as DFU, due to the weak adhesion to wet tissue (for fibrin glue) or poor biodegradability (for cyanoacrylate). By contrast, d-SMG is biodegradable with both strong adhesion and good biocompatibility. As a nourishing food, the global production of the *A. fulica* snail has reached to about 40,000 tons per year[49]. Thus, the snail secretion d-SMG is a renewable and sustainable resource, and its large-scale production may be feasible.

As a natural biomaterial consisting of 30–50% protein and 10–16% sulfated glycosaminoglycan, d-SMG has a composition similar to that of the extracellular matrix (ECM), which is composed of fibrous proteins and polysaccharides, and provides mechanical support for cell growth and regulates cell behavior[50]. In recent decades, ECM-like wound dressings have been developed as provisional matrix replacements for diabetic wounds[51]. These products allow host cellular infiltration and promote matrix deposition, angiogenesis, and re-epithelialization[51]. An FDA-approved acellular wound dressing (Integra Dermal Regeneration Template, Integra Life Sciences) comprises an inner layer of collagen and GAG and an outer silicone layer, which benefits complete wound healing[38,52,53]. Likewise, the s-GAG-protein-based gel d-SMG can keep the wound tissue moist and provide a suitable environment for tissue regeneration. In vivo study showed that d-SMG promoted skin wound healing in both normal and diabetic rat models (Supplementary Fig. 11, Fig. 6). It improved the angiogenesis, granulation tissue neogenesis, collagen deposition, and epidermal regeneration in the wound bed (Fig. 9c). Apart from functioning as an adhesive wound dressing, d-SMG also reduced the levels of inflammatory cytokines in wound tissues, possibly by both regulating macrophage polarization and capturing inflammatory cytokines.

Normal wounds heal via predictable, sequential steps: hemostasis, inflammation, proliferation, and remodeling, whereas chronic wounds, especially DFUs, exhibit delayed healing and are often considered to be 'stuck' in the inflammatory stage[54]. Macrophages are the main cells in regulating the wound healing process by differentiating into the proinflammatory M1 and anti-inflammatory M2 phenotypes in different phases[17]. Both d-SMG dressing and its active ingredient s-GAG could promote the polarization of macrophages towards the M2 phenotype possibly by upregulating STAT3 phosphorylation, which may contribute to transition of wound healing process from inflammation to proliferation, thus accelerating wound healing (Fig. 6). The s-GAG, as the main active component of d-SMG, is responsible for the inflammation modulating function of d-SMG, while s-GAG itself does not have potent adhesion.

It's reported that negative-charged glycosaminoglycan can efficiently bind to the inflammatory cytokines with positive-charged amino acids[43]. For instance, the well-known anticoagulant heparin can bind to inflammatory cytokines, and exhibits potent anti-inflammatory function[55–57]. A heparin-based star-shaped polyethylene glycol hydrogel system was designed to capture inflammatory chemokines, which showed in vivo pro-healing effect[40]. d-SMG contains a high content of heparin-like GAG, which also exhibits high affinity for binding the inflammatory cytokines such as TNF-α, IL6, IL8, and IP10. The result suggests that d-SMG may capture inflammatory cytokines in wound tissues.

For the research of DFU, although consensus is lacking regarding the animal model that best recapitulate human DFU pathophysiology, streptozotocin (STZ)-induced diabetic model is one of the most applied models to evaluate diabetic wound healing functions of biomaterials[58]. However, this model has limits in mimicking human DFU in terms of pathophysiology, for instance it may not have the similar peripheral neuropathy, prolonged systemic hyperglycemia and mechanical stress etc[38]. Nevertheless, our data have exhibited optimistic results for the concept of using natural adhesive in wound healing. Further assessment of d-SMG application may require animal models with delayed wound healing such as db/db mice and porcine, which are more closely resembles the physiology of human skin to illustrate the translational and clinical potential[35,59].

In this work, d-SMG from snail secretion is a natural tissue adhesive, which consists of a network of protein and sulfated polysaccharide. The biodegradable d-SMG have strong adhesion to wet tissue, hemostatic effect, good biocompatibility and hemocompatibility, and in vivo pro-healing activity for skin wound. Sulfated glycosaminoglycan is the main active components in d-SMG, which can effectively regulate the inflammatory response in wound microenvironment by promoting the polarization of macrophages towards the anti-inflammatory M2 phenotype, and by capturing inflammatory cytokines. Our findings provide theoretical and material insights into developing bioactive dressings and bioengineered scaffolds for wound healing.

## Methods
### Materials
1, 9-dimethyl methylene blue and streptozotocin (STZ) were purchased from Sigma-Aldrich (MO, USA). L-DOPA was from Shanghai Yuanye (Shanghai, China). 3-(4,5-dimethyl-2-thiazolyl)–2,5-diphenyl tetrazolium bromide (MTT), BCA protein assay kit, and alizarin red S were from Beijing Solarbio (Beijing, China). Absorbable sutures were from Shanghai Pudong Jinhuan Medical Products. (Shanghai, China). Cyanoacrylate (COMPONT®) was from the Beijing Compete Medical Equipment (Beijing, China). Fibrin glue (Bioseal®) was from Guangzhou Bioseal (Guangzhou, China). Nonabsorbable surgical sutures were from Foosin Medical Supplies Inc., Ltd. (Weihai, China). The Alginate dressing™ was from Minnesota Mining Manufacturing (3 M) (St. Paul, Minnesota, USA). All cell culture-related reagents were from Gibco (Grand Island, NY, USA). All other chemical reagents are of analytical grade from commercial sources.

### Animals and cells
The snail *Achatina fulica* (20–23 g/snail) was obtained from the Fangyuan Snail Farm (Xiangyang, China), and *Helix lucorum* (15–19 g/snail) was from Lver Agricultural Science and Technology Park (Luoyang, China). Male SD rats (8–12-week-old) were from Hunan Shrek Jingda Experimental Animal Co., Ltd. (Hunan, China). The animal experiments were approved by the Research Ethics Committee of the Kunming Institute of Botany (SYXK-K2018-0005), Chinese Academy of Sciences.

L929 (GDC0034) cell line was purchased from China Center for Type Culture Collection, Chinese Academy of Sciences. RAW264.7 (SCSP-5036) cell line was purchased from National Collection of

Authenticated Cell Cultures, Chinese Academy of Sciences. Cell lines were authenticated by providers. All cell lines showed normal size, karyotype, morphology, and no contamination were observed.

## Collection of SMG and preparation of d-SMG

The snails were cleaned with deionized water and sterilized in ozone atmosphere for 30 min. By stimulating snail foot, the fresh snail-mucus gel (SMG) was collected from the secretion of snails. Approximately 20 g of fresh SMG can be collected from 10 adult snails each time, and the snails were bred for another two weeks before next collection. The dried SMG (d-SMG) was obtained from SMG by lyophilization (SMG was placed in liquid nitrogen for 15 min, and then placed in a lyophilizer at a pressure lower than 75 Pa for 48 h). Then, after irradiation the sterilized d-SMG was sealed and stored at −20 °C before being used in animal experiments.

## Extraction and purification of s-GAG

The s-GAG was isolated and purified as described previously with some modifications[18]. 10.8 g d-SMG was added to 800 mL deionized water containing 2% alkaline protease, after incubation at 60 °C and pH=9 (maintained with 6 M NaOH) for 36 h, the mixture was cooled to room temperature, pH was adjusted to 7 with 6 M HCl, and the solution was diluted to 1000 mL with deionized water. Then after centrifugation (3000 ×g, 30 min), the supernatant was precipitated with 3 volumes of ethanol, the obtained precipitate was washed with absolute alcohol, dissolved in 100 mL of 0.2 M NaCl and centrifuged at 3000 ×g for 20 min. 5 % benzethonium chloride solution was added to the supernatant and after centrifugation the precipitate was dissolved in saturated NaCl, precipitated with 4 volumes of absolute alcohol. The precipitate was dissolved in water, dialyzed with 3 kDa cut-off membrane, and lyophilized to obtain a white powder designated s-GAG (1.2 g).

## Characterization

Rheological tests were performed using a rheometer (Anton Paar MCR302, Germany) with rheocompass software. Fresh SMG (0.5 g) and rehydrated d-SMG (with hydration rate of 95%, 90%, 80%, and 60%) were prepared and placed on a parallel plate. Subsequently, the storage modulus (G′) and loss modulus (G″) were recorded with frequency of 0.1 to 1 Hz and a shear strain of 5% at 25 °C.

The materials and methods of the chemical composition analysis are described in detail in the Supporting Information.

## In vitro adhesion tests

Qualitative evaluations of tissue adhesion were performed using rat tissues or organs, including the heart, liver, kidney, spleen, and muscle. Approximately 200 mg of fresh mucus and 5 mg of d-SMG was placed on the wet tissue surfaces, and the adhesion state was observed and photographed.

Quantitative tissue adhesion strength tests, including the lap shear, tensile, and T-peel tests, were performed according to the ASTM F2255-05, F2258-05, and F2256-05. Commercial hog casings and skins were used as biological tissue materials. Tissues were coated with water or blood before testing, and then d-SMG was placed between the two tissues and secured for 1 min. The two tissues were placed into a universal testing machine (Yueqing HANDPI Instrument, China) for tensile loading at a strain rate of 1 mm/s. The hydrogel adhesion strength was determined at the point of detachment.

## Cytocompatibility

The cytotoxicity of d-SMGs was evaluated as previously reported[27] with minor modifications. Briefly, d-SMG was dissolved in complete medium. The mixed medium was centrifuged (22000 × g, 15 min), and the supernatant was filtered through a 0.22-μm membrane. L929

fibroblast cells were seeded into 96-well plates (2000 cells per well) and cultured at 37 °C for 24 h. Then, the medium was replaced with the normal medium or mixed medium (0.05, 0.1, 0.2, 0.5, 1.0, and 2.0 mg/mL d-SMG) and the cells were continuously cultured at 37 °C for 24, 48, and 72 h. At selected times, the culture solution was replaced with an MTT reagent for 4 h at 37 °C. Next, MTT was gently removed, 100 μL of DMSO was added to each well, and the absorbance at 490 nm was detected. Three batches of L929 cells were used for biological repeats. At the same time, the cell growth status was recorded in real time by IncuCyte S3 Live-Cell Analysis System (Sartorius, Germany). The number of cells in each well was counted by IncuCyte Cell-By-Cell analysis module.

## Hemocompatibility

The hemocompatibility test was performed as previously described[60]. Anticoagulated blood from healthy SD rats was diluted to a concentration of 5.0% (v/v) with PBS. 0.5 mL d-SMG (0.1, 0.2, 0.4 mg/mL in PBS) was added into 0.5 mL whole blood diluent (5.0% in PBS). After incubation at 37.0 °C for 1.0 h, the samples were centrifuged (116.0 × g, 10 min), and the absorbance of the clear supernatant was detected at 540.0 nm using a microplate reader (FlexStation 3 Multi-Mode Microplate Reader, Molecular Devices, LLC, USA). Blood diluted to 5.0% with deionized water and PBS was served as the positive control and negative control, respectively. The hemolysis percentage of d-SMG was calculated as follows:

$$\text{Haemolysis ratio (\%)} = \frac{A_{\text{sample}} - A_{\text{control}}}{A_{\text{positive}} - A_{\text{control}}} \times 100\% \tag{1}$$

where $A_{\text{sample}}$, $A_{\text{control}}$, and $A_{\text{positive}}$ represent the absorbance of the sample, negative control, and positive control, respectively.

APTT was determined using a coagulometer (TECOMC-4000, Germany) using APTT reagents and human coagulation control plasma, according to the product instruction. Heparin was used as the positive control.

## Biodegradability

The biodegradability of the d-SMG was evaluated by subcutaneous embedding in rats. Male SD rats (200 ± 20 g) were randomly divided into three groups (n ≥ 5/group). After anesthesia with isoflurane, the dorsal area was shaved and disinfected with iodine and 75% ethanol. Dorsal skin (3.5 cm from the ear) was incised with the length of 2 cm. Skin incision were sutured and bandaged after embedding 20 mg of d-SMG to prevent scratching. Each rat was housed in a single cage and allowed *ad libitum* access to food and water. Photographs of the wound were taken on day 0, 7, 14, and 21. Square tissues (3 × 3 cm around the scar) were collected for histological analysis on day 3, 7, 14, and 21.

## BCI test

The BCI test was performed as previously described[22,23]. Whole blood was collected from healthy rats through the abdominal aorta and was anticoagulated. d-SMG was cut into square slices (length: 6 mm; height: 1 mm). Gauze was used as a positive control and cut to the same size as the hydrogel slice. The dried hydrogel or gauze was preheated at 37 °C. Then, 10.0 μL of recalcified whole blood (contained 10 μM CaCl₂) was incubated with the dried hydrogel or gauze to initiate coagulation. After incubation at 37 °C for 60, 90, 120, and 150 s, 1.0 mL deionized water was gently added to lyse the free red blood cells. After centrifugation (116.0 × g, 10 min), the absorbance of hemoglobin in the supernatant was detected at 540.0 nm using a microplate reader. BCI

was calculated by the following equation:

$$BCI\,(\%) = \frac{A_{\text{sample}}}{A_{\text{control}}} \times 100\% \qquad (2)$$

where $A_{\text{sample}}$ and $A_{\text{control}}$ represent the absorbance of the sample and control (10.0 μL recalcified whole blood dissolved in 1.0 mL deionized water), respectively.

### Red blood cell and platelet adhesion assessments
d-SMG or gauze was incubated with whole blood for 5 min or with platelet-rich plasma for 1 h at 37 °C. After incubation, all samples were washed thrice with 4 mL DPBS to remove free blood cells or platelets. The samples were then fixed in 3% glutaraldehyde for 4 h, gradient-dehydrated with ethanol, sputter-coated with platinum, and examined using scanning electron microscopy (Sigma 300, Zeiss, Germany), (Magnification: 800× and 2000 ×, Power: 7.00 kV).

### In vivo liver hemostatic ability
Male SD rats (200 ± 20 g) were used for liver hemostatic tests in vivo. The rats were randomly divided into four groups: control, *A. fulica* d-SMG, *H. lucorum* d-SMG, and the gauze (positive control). Rat was anesthetized with isoflurane, and liver was exposed by abdominal incision. The pre-weighed filter paper was placed beneath the liver. A part of the left lobe of the liver (length: 2.5 cm, width: 0.5 cm) was cut off, and hemostatic material was immediately applied to the vertical section. After 2 min, the filter paper was weighed, and blood loss was calculated. In the control group, no treatment was applied after the liver was cut.

### Skin incision adhesion assessments in vivo
A rat skin incision model was used to evaluate the in vivo tissue adhesion of d-SMG, using sutures, cyanoacrylate, and fibrin glue as positive controls and saline treatment as a negative control[27]. Rats were divided into five groups (7/group), rat dorsal was shaved 12 h before experiment. After anesthesia with isoflurane, and two linear skin incisions (2 cm in length) were made in the rat dorsum using sterilized surgical scissors. Then the skin incision was photographed immediately after treatment, and covered with an elastic bandage. Each rat was housed in a cage and allowed *ad libitum* access to food. Wound closure was recorded on postoperative day 3, 5, and 7. Circular skin samples (3 × 3 cm around the wound) were collected for histological analyses on day 7, after which the rats were euthanized.

### Wound healing assessment in normal rats
SD Rats (male, 180–220 g) were randomly assigned to four groups (8–10/group): the negative control (normal saline), *A. fulica* d-SMG (5 mg in 50 μL of normal saline), *A. fulica* s-GAG (10 mg/mL, 90 μL), and the positive control Alg™ dressing (10–12 mm in diameter) groups. Rats were anesthetized with isoflurane, and the dorsal area was shaved and disinfected with iodine and 75% ethanol. A 10-mm circular, full-thickness dorsal wound was created by a punch, then skin around the wound was fixed with a gasket. The Alg™ dressing was changed after obvious absorption of exudate, while the degradable d-SMG was not changed during experiment. Each rat was housed in a single cage and allowed *ad libitum* access to food and water. The wound was photographed on day 0, 2, 5, 8, and 11. Circular skin samples (3 × 3 cm around the wound) were collected for histological analyses on day 5 and 15, after which the rats were euthanized.

### Wound healing assessment in diabetic rats
The chronic wound healing ability was evaluated by the full-thickness skin wound model in streptozotocin (STZ)-induced diabetic rat. Diabetic rats were established as previously described, then the SD rats (male, 320 ± 30 g) were divided into four groups (≥13–16/group). The rats were anesthetized with isoflurane, and their dorsum was shaved and disinfected with iodine and 75% ethanol. A 10-mm circular, full-thickness dorsal wound was created in each animal using a punch. The wound treatment is the same as those of normal rats. Each rat was housed in a single cage and allowed *ad libitum* access to food and water. The wound was photographed on day 0, 3, 7, 10, 14, and 21. Circular skin tissue from three randomly selected rats (3 × 3 cm around the wound) were collected for histological analysis, after which the rats were euthanized.

### Histological analysis and immunofluorescence staining
For histopathological analysis, wound tissues from diabetic rats were collected on day 3, 7, and 14, fixed with 4% paraformaldehyde, embedded in paraffin, and sectioned into 5-μm thick sections. Tissue sections attached to the slides were deparaffinized, rehydrated, and stained with hematoxylin & eosin, and Masson's trichrome.

For immunofluorescence staining, deparaffinized and rehydrated sections were processed for heat-induced antigen retrieval in citrate buffer (10 mM, pH 6.0) at 98 °C for 10 min. Non-specific binding was blocked by goat serum (10%) for 1 h after permeation for 10 min. To evaluate angiogenesis, CD31 and α-SMA primary antibodies and the corresponding secondary antibodies were used. CD206 and CD86 primary antibodies and corresponding secondary antibodies were used to determine macrophage phenotypes in the wound tissue. Cellular nuclei were counterstained with 4',6-diami-dino-2-phenylindole (DAPI, Servicebio, China) for 10 min. Immuno-fluorescence images were acquired using a Carl Zeiss Microscopy GmbH (Axio Scan Z1, Germany) and quantified using ImageJ software.

### Cytokines analysis by bio-plex system
For cytokine assays, fresh skin wound tissues were obtained from diabetic rats. Subsequently, the tissues were homogenized in cold PBS using a homogenizer (KZ-III, Servicebio, China) with a rotor-stator (4 mm beads × 3, 3 mm beads × 4, 60 s × 2 replications at 60 Hz), and then centrifuged at 15,000 × $g$ at 4 °C for 10 min. Protein concentration in supernatant was determined using a BCA protein assay kit. The concentrations of interleukin-1a (IL-1a), interleukin-1b (IL-1b), interleukin-2 (IL-2), interleukin-4 (IL-4), interleukin-5 (IL-5), interleukin-6 (IL-6), interleukin-10 (IL-10), interleukin-12 (IL-12), interleukin-13 (IL-13), granulocyte-macrophage colony stimulating factor (GM-CSF), interferon-gamma (IFN-γ), and tumor necrosis factor-α (TNF-α) were measured by the flexible Bio-Plex system (Bio-Rad, #171K1002M, USA). All procedures were performed according to the manufacturer's instructions.

### Surface plasmon resonance
Interactions between the s-GAG or heparin and cytokines were assessed by detecting the binding of s-GAG or heparin to TNF-α, IL-8, IP-10, and IL-6 coated on CM5-Chip (Cytiva, #10308493, USA) using surface plasmon resonance (Biacore S200, GE Healthcare), as previously described[18]. Briefly, cytokines (TNF-α (#ab25010), IL-8 (#ab51095), IP-10 (#ab9810), and IL-6 (#ab259381), Abcam, UK) was respectively coupled to a CM5 sensor chip of a Biacore S200 via carbodiimide chemistry at pH 5 in 10 mM acetate buffer using EDC/NHS (Aladdin, Shanghai, China) to a level of 1000RU. The s-GAG or heparin was then run over the surfaces at 30 μL/min flow rate at different concentrations (10.0, 8.0, 4.0, 2.0, 1.0, 0.5, and 0.25 mg/mL for s-GAG to TNF-α and IP-10; 6.0, 3.0, 2.0, 1.0, 0.5, and 0.25 mg/mL for s-GAG to IL-6; 7.0, 5.0, 4.0, 3.0, 2.0, and 1.0 mg/mL for s-GAG to IL-8; 3.0, 2.0, 1.0, 0.5, 0.25, 0.125, and 0.063 mg/mL for heparin to TNF-α, IL-8, IP-10, and IL-6) in duplicate and from the obtained sensor grams, the kinetic association and dissociation rate constants were determined by curve fitting using the Biacore S200 Evaluation Software.

The kinetic curve of interactions between the s-GAG or heparin and cytokines was fitted by using the Biacore S200 Evaluation Software.

## Western blot analysis

RAW 264.7 cells treated with d-SMG (0.2 mg/ml), s-GAG (0.15 mg/ml), IFN-γ (10 ng/ml), IL-4 (5 ng/ml), or IL-10 (20 ng/ml) for 2.0 h were lysed using RIPA buffer with 1% phosphatase and proteinase inhibitor (Beyotime, China). Following centrifugation of the homogenate ($20,000 \times g$, 15 min), the supernatant was collected for western blotting. Protein concentrations were measured using the BCA protein assay (Beyotime, China), with bovine serum albumin (BSA) as a standard. Protein samples (20 μg/lane) were loaded onto a 10% SDS-PAGE gel for separation and then transferred to a polyvinylidene difluoride (PVDF) membrane. Membranes were blocked in 5% non-fat milk (Bio-Rad) for 30 min at room temperature and then incubated with antibodies against STAT1, p-STAT1 (Tyr701), STAT3, p-STAT3 (Tyr705), STAT6, p-STAT6 (Tyr641), and β-Actin at 4 °C overnight (Supplementary Table 11). Blots were then washed in PBST (PBS with 0.1% Tween-20) and incubated with secondary antibody. Immunoreactive proteins were detected using an ECL chemiluminescence system (Clinx Science Instruments Co., Ltd., Shanghai, China) with default settings. The ratio between phosphorylated protein and total protein was calculated.

## Transcriptome analysis

Wound tissues from diabetic rats were collected on day 3, 7, and 14. Total RNA was extracted using TRIzol reagent (Invitrogen). RNA was detected by examining the A260/A280 ratio using a NanodropTM One spectrophotometer (Thermo Fisher Scientific Inc., USA), and quantified using Qubit3.0 with a QubitTM RNA Broad Range Assay kit (Thermo Fisher Scientific Inc., USA). 2 mg of total RNA were used for stranded RNA sequencing library preparation using the Ribo-off rRNA Depletion Kit (Human/Mouse/Rat) (Illumina, USA) and the KCTM Stranded mRNA Library Prep Kit for Illumina (Wuhan Seq-Health Co., Ltd. China), according to the manufacturer's instructions. The library products corresponding to 200–500 bp were enriched, quantified, and sequenced on a NovaSeq 6000 sequencer (Illumina, USA) with the PE150 model. RNA-seq data were compared using bioinformatic tools. Differential expression analysis was performed using the edgeR software[61]. Heatmaps were generated using the TBtools software (https://github.com/CJ-Chen/TBtools/releases). GO terms, and KEGG pathways were identified using KOBAS 2.0. Hypergeometric tests and Benjamini–Hochberg FDR control procedures were used to determine the enrichment of each term.

## mRNA expression analysis by qRT-PCR

Total mRNA was extracted from wound tissue or RAW264.7 using an E.Z.N.A. Total RNA Kit II (Omega Biotek, USA) and reverse-transcribed into cDNA using BlasTaqTM 2X PCR MasterMix (Abm, USA), according to the manufacturer's instructions. qRT-PCR was performed using BlasTaqTM 2X PCR MasterMix (Abm, USA) on a QuantStudioTM 7 Flex Real-Time PCR detection system (Thermo Fisher, USA). Levels of *Mrc1, Arg1, Cd163, Jak3, Il10, Nos2, Tnf, IL-1b, Cd86, Tgf-b, Vegf, Col1a1, Col3a1, Pdgf*, and *β-actin2* transcripts were examined. The data were analyzed using the $2^{-\Delta\Delta Ct}$ method, and the expression was normalized to that of β-actin. The experiments were independently performed three times ($n = 3$). The mRNA primer sequences used are listed in Supplementary Table 12.

## Statistical analyses

All data are presented as the mean ± standard error of the mean (SEM) and were analyzed by Student's two-tailed $t$ test. $P$-values less than 0.05 were considered statistically significant. The number of independent samples of microscopic images of pathological sections is consistent with that of corresponding quantitative experiment. SPSS software (version 20.0; IBM, USA) and OriginPro 2017 SR2 (b9.4.2.380) were used for the data analysis and plotting.

## Reporting summary

Further information on research design is available in the Nature Portfolio Reporting Summary linked to this article.

## Data availability

All data needed to support the conclusions in the paper are present in the paper and/or the Supplementary Information. Data underlying Figs. 1–9 and Supplementary Figs 1–17 is provided with this paper in the Source Data file. The mass spectrometry proteomics data have been deposited to the ProteomeXchange Consortium with the dataset identifier PXD036781. The transcriptome sequencing data used in this study are available in the Gene Expression Omnibus database under accession number GSE206113. Source data are provided with this paper.

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

## Acknowledgements

This work was supported in part by the National Youth Talent Support Program to M.Y.W., the Youth Innovation Promotion Association CAS (Y2021104 to M.Y.W.), CAS "Light of West China" Program to M.Y.W. and the Ten-thousand Talents Program of Yunnan Province (YNWR-QNBJ-2018-271 to M.Y.W.), and Yunnan Fundamental Research Projects (grants

202201AS070073 to M.Y.W. and 202101AT070152 to Z.P.Z.). We appreciate Mr. Yong Tian and Mr. Yuncai Tian from the Shanghai Zhenchen Cosmetics Co., Ltd. We thank the Yunnan Institute of Medical Device Testing for the rheological testing. We also thank Mr. Zhijia Gu for the scanning electron microscopy imaging.

## Author contributions

M.Y.W. and T.D. conceived the idea for the d-SMG adhesive. T.D., Z.X.J., L.L., L.Y., and A.K.Z. conducted the in vitro and ex vivo experiment and analysis. T.D., D.X.G., X.M.S, L.X.Z, Z.P.Z., and M.X.T. designed and conducted the in vivo rat studies and analysis., L.H. and H.B.Q. supervised the study. M.Y.W. conceived the project, designed the experiments, supervised the study and wrote the paper. All authors reviewed and provided input to the manuscript.

## Competing interests

A patent has been submitted in part entailing this work. The patent authors contain M.Y.W. and T.D., the institution is "Shanghai Zhenchen Cosmetics Co., Ltd.", and the patent application number: CN202210212225.8. The patent title is "A biomaterial and its preparation method and application". This patent relates to the field of biotechnology. The patent status is substantive examination. The remaining authors declare that they have no other competing interests.
