## [Peer Review File · Nature Communications]

A Natural Biological Adhesive from Snail Mucus for Wound RepairEditorial Note: Any mention of referees' names has been redacted due to referees' wishes.

REVIEWER COMMENTS

Reviewer #1 (Remarks to the Author):

This manuscript describes the use of unpurified snail mucus as a tissue adhesive and a wound dressing. The authors demonstrate well the remarkable adhesive and wound healing properties of these materials, extracted from two different snail species. The manuscript presents a large set of data, ranging for the chemical and material characterization of the secretions, up to their in vivo evaluation in two wound models. The manuscript is a great example of how new biomaterial can emerge from looking at Nature. An although the science is of excellent quality, there are some shortcomings in the presentation of the data and in its interpretation and discussion that should be addressed before publication.

1. My first impression is that the authors could have gone a bit deeper into the material characterization given the large amount of work that has been performed to test its application.
 - For instance, how important is calcium? This could be explored by dialysis and or adding chelators.
 - More information about interactions between the components and the tissue could have been obtained with Raman spectroscopy.
 - The main proteins could have been identified by mass spectrometry.
2. The authors provide a molecular weight based on the SDS-PAGE analysis of the soluble fractions of both secretions. This seems to have been the only analysis done in this way. It is unclear what proportion of the material is soluble or insoluble. What is the role of the insoluble fraction?
3. Could the authors comment on the use of fibrin as an adhesion comparison group. How would other approved tissue glues perform compared to the d-SMG? I see that the authors have included a cyanoacrylate comparison for Figure 5 which could be interesting to look at in terms of adhesion.
4. Figure 1. The SEM image were of freeze-dried samples. This method of dehydration is known to form large pores into the materials due to the formation of ice crystals. The representatives SEM images thus do not bring a lot of information.
5. Figure 1j, k. Could the authors write in the figure legend that the percentages are hydration%. I had to look it up elsewhere, Also, it would be good to understand what the % hydration of the fresh samples are in comparison (relates back to my previous question on d-SMG vs fresh-SMG).
6. I am sure the authors say it somewhere, it is not clear what hydration was used for all the evaluations passed the rheology. Please indicate that in the text, and ideally in the name of the d-SMG in all subsequent figures.
7. Page 4 – line 143. It is not clear why the authors think that the d-SMGs are stronger than their fresh counterparts. Is that for an equivalent hydration rate?
8. Page 4 line 153 – The authors mention 5% hemolysis as a threshold for biocompatibility. Please add a reference to back that up.
9. Figure 3. Regarding the cytocompatibility data. The s-SMG show increase in metabolic activity that seems to go beyond experimental variations. This could be linked with increase proliferation or metabolic activity. Could the authors look into this and discuss?
10. Please explain what concentration is used for the cytocompatibility assay. It is not mentioned on page 20 in the methods either. That seems like an important information.
11. Figure 6. Regarding the effect of d-SMG on would healing. I might have missed it, but it is not

clear from the manuscript what is the GAG that was used as a control in the wound healing studies (Figures 6 and 7). This is especially important to know given the good results the GAG conditions showed in these studies compared to the d-SMG

12. It is a bit surprising to me to see that although the diabetic mice model does provide slower wound closure time than the healthy mice, wound closure in untreated animals is still relatively rapid. Would this be considered chronic wound? Could the authors comments on the limits of this model in the context of this material testing?

13. It seems like the angiogenesis markers are indeed upregulated in the diabetic wound model for d-SMG and GAGs. But since the authors have nice immunohistochemistry done on these tissues, would it not also make sense to directly measure the number of vessels and their size as a more direct proof of angiogenesis?

14. Figure 7 Regarding inflammatory regulation. The effect of GAG and Alginate is strong in the immune regulation. The authors have not commented or discussed this result in depth. How does it compare to the literature? It was not my impression that alginates could have such a strong immune dampening effect.

15. The authors have not shown the effect of Alginate and GAG on the macrophage cell line Figure 7f. This would be interesting to support the in vivo results.

16. The authors mention that the GAG fraction in the d-SMG could capture inflammatory cytokines and participate in the immune modulatory effect. Could the authors discuss more their data in that context? Do the authors think that the low cytokines levels measured are due to their sequestering, which prevent them from being sampled? Or are their biological effects leading to the inhibition of their secretions? How does the in vitro assay on macrophage inform you in this regard?

17. Could the authors also enrich the discussion by describing what they see for this material. Can scaleup of production be achieved? In that case, can sterilization be achieved without affecting the properties? Would the next step be to further dissect the material composition and create synthetic versions?

Minor issues:

Figure 4e. That graph has a different style than others, it would be good to harmonize

Page 4 line 179 – “The control group had...” Please just say what that group was. Is it fibrin?

Page 3, line 123. Reference 21 seems a bit misplaced here. Is heparin really secreted in intestinal mucus or is it part of the tissue

Page 6, section on Inflammatory regulation line 267. It is unclear why IL-1Beta is mentioned before and separately than the other inflammatory markers mentioned in the next phrases.

Figures 5c, 6c, 6g indicate “rates” on the y-axis. These values are not technically rates since they are provided for a given time, and not relative to time.

Figure 7a. The conditions are not presented in the same order as on other graphs. Please keep the order the same.

Figure 8c, 8d. It is not clear what the numbers on the scale are.

Figure 8e, 8f, what in “cound”. Is “rich factor” a typo. Did you mean “enrichment factor”

Page 21, line 599. D-SMG instead of SMG. Please indicate the controls used in these experiments.

Reviewer #2 (Remarks to the Author):

REFEREE'S RECOMMENDATIONS

Accept with major changes.

Comments for the Authors:

The study contains clear and promising data and well written. However, I have some suggestions as below.

Line 218 & line 219

Authors must add *A. fulica* before d-SMG group in both lines. Also, the same comment on caption of figure 11 in supplementary data.

Line 302

Authors must add magnification power of scanning microscope.

Line 305

Revise it. I think it may be for l not for c.

Page 10 Fig 3d

Word day must be capitalized.

Page 14 Fig 7

Why authors didn't put results of day 14 as mentioned in material??? It must be clarified.

Page 14 Fig 7 caption

Which day did you mean???? Day 3 or 7 or 14. Revise it

Line 492

Achatina fulica and *Helix lucorum* must be italic.

Line 508

Why authors write fresh SMG or rehydrated??? Which one do you use exactly??? It must be clarified.

Page 19

Authors must add separation and purification method of GAG from d- SMG after characterization or in supplementary material after chemical composition analysis.

Line 515

Authors must write and instead of or.

Line 533

Authors must mention effect of GAG of 2 snails on human coagulation plasma in material as in results. It must be clarified.

Line 551

ad libitum must be italic.

Line 575

Authors must write name of apparatus and its company of scanning microscope.

Line 577& 578

in vivo, *Achatina fulica* and *Helix lucorum* must be italic. Please revise.

Line 585& 591

in vivo and *ad libitum* must be italic.

Line 599& 610

Revise, authors use SMG or D- SMG as in results.

Page 22 line 604

In this section, why authors didn't use GAG in wound healing in normal rats, also??? Why in diabetic only???

FIELD OF EXPERTISE OF REFEREE: Microbiology

Reviewer #3 (Remarks to the Author):

This manuscript reported the efficacy of snail mucus to act as a bio-adhesive material, to be used in wound repair.

The results are quite novel. There are already papers on the wound repair activity of snail mucus. The novelty of this research is mainly associated with the creation of an adhesive matrix.

The need for innovative products to repair epithelial damages makes this research of interest, even if already several data are present in the literature (The Effectiveness of Snail Slime and Chitosan in Wound Healing. Agnes Sri Harti et al. International Journal of Pharma Medicine and Biological Sciences Vol. 5, No. 1, January 2016; Application of snail mucin dispersed in detarium gum gel in wound healing. Michael Adikwu et al. Scientific Research and Essays 2(6):195-198, 2007; The Protective Effect of Snail Secretion Filtrate in an Experimental Model of Excisional Wounds in Mice. Enrico Gugliandolo et al. Vet Sci

. 2021 Aug 20;8(8):167. doi: 10.3390/vetsci8080167.; HelixComplex snail mucus exhibits pro-survival, proliferative and pro-migration effects on mammalian fibroblasts. Trapella Claudio et al. Sci Rep . 2018 Dec 5;8(1):17665. doi: 10.1038/s41598-018-35816-3.).

The research is well performed but some concerns are present:

- 1) The preparation of snail mucus is not clearly stated. It is important to clarify if any filtration is performed.
- 2) No data on microbiological content of snail mucus is reported. As it is used on injured skin, it is necessary to prove the sterility of the product.
- 3) It is mandatory to present a table indicating the content of the snail mucus. It is important to comment on the possible differences between *Achatina* and *Helix lucorum* mucus.
- 4) Do the authors have standardized the snail mucus collection? Are all the batches comparable in the present compounds?

This is an important point to be sure of the standardized efficacy of the final product. This is important also in order to make the research to be reproducible.

Taking into consideration the above-mentioned concerns, the manuscript needs a careful revision, taking into consideration the additional information needed.

VIEWER COMMENTS

Our Response: The reviewers' work is greatly appreciated. The following are point-by-point responses to the reviewers' comments, and according to their suggestions, we have revised the manuscript, with changes marked red in the resubmitted text.

Reviewer #1 (Remarks to the Author):

This manuscript describes the use of unpurified snail mucus as a tissue adhesive and a wound dressing. The authors demonstrate well the remarkable adhesive and wound healing properties of these materials, extracted from two different snail species. The manuscript presents a large set of data, ranging for the chemical and material characterization of the secretions, up to their in vivo evaluation in two wound models. The manuscript is a great example of how new biomaterial can emerge from looking at Nature. Although the science is of excellent quality, there are some shortcomings in the presentation of the data and in its interpretation and discussion that should be addressed before publication.

Our Response: We are thankful to the reviewer for his/her positive comments on our work and for his/her important contributions to improve the manuscript.

1. My first impression is that the authors could have gone a bit deeper into the material characterization given the large amount of work that has been performed to test its application.

Our Response: Thanks for the reviewer's instructive advice on the material characterization of d-SMGs.

- For instance, how important is calcium? This could be explored by dialysis and or adding chelators.

Our Response and Revision: In order to explore the role of calcium, we tried both dialysis and chelation for removal. The d-SMG solution was dialyzed with a dialysis bag (500 Da cut-off) in deionized water for 4 days, then lyophilized. The content of calcium was reduced by 60~70%, while the sodium was reduced by over 99% (Supplementary Table 8). Additionally, after dialysis in the presence of 1% EDTA-2Na for 4 days, the calcium content in d-SMG was reduced by about 99%. Next, we evaluated adhesion strengths of d-SMG before and after dialysis using a Lap-shear test. The adhesion strength of d-SMG was not observed to obvious change after dialysis (Supplementary Figure 8b). These results indicated that calcium ions may play a little role for the tissue adhesion of the snail-derived biomaterial d-SMG.

We have provided this information in the resubmitted manuscript in Supplementary Figure 8b, Table 8 and Results section.

Supplementary Table 8. Content of Ca²⁺ and Na⁺ in d-SMG under different treatments.

	Before dialysis		After dialysis in water		After dialysis in 1% EDTA solution	
	A. fulica d-SMG	H. lucorum d-SMG	A. fulica d-SMG	H. lucorum d-SMG	A. fulica d-SMG	H. lucorum d-SMG
Ca ²⁺ (mg/g)	26.4	66.7	10.8	18.6	0.31	0.44
Na ⁺ (mg/g)	63.1	42.4	0.41	0.30	0.68	0.69

Supplementary Fig. 8b. Adhesion strength of d-SMG before and after dialysis.

Data are presented as the mean \pm SEM, $n = 3$

- More information about interactions between the components and the tissue could have been obtained with Raman spectroscopy.

Our Response and Revision: Thanks for the reviewer's good suggestion. We performed Raman mapping to obtain more information about interactions between d-SMG and the tissue. An obvious diffusion region of about 300 nm in the interface between d-SMG and tissue was observed, owing to the remarkable difference of response value of Raman scattering on d-SMG and tissue at 565 cm^{-1} (disulfide bond), 1090 cm^{-1} (aromatic ring and sulphonic acid), 1650 cm^{-1} (amide I), and 2930 cm^{-1} (C-H bond) (Smith, W.E. and Dent, G. *Modern Raman Spectroscopy – A Practical Approach*. pp. 15-20, John Wiley & Sons, Inc., Hoboken, 2005). These results indicated the mutual penetration between d-SMG and tissues. Moreover, there are disulfide bond, hydrogen bonds, π - π interactions, electrostatic interaction, and hydrophobic interactions between d-SMG and tissues, which lead to a strong adhesion between the d-SMG and the tissue. We have provided these data in Supplementary Figure 7 and Results section in the resubmitted manuscript.

Supplementary Fig. 7. Raman spectroscopy analysis of the adhesion interface between d-SMG and tissues a and c Raman spectrum scanning area of d-SMG, interface, and tissue. b and d Raman mapping images of the d-SMG, tissue, and their interface. e and f Raman spectra of the d-SMG, tissue, and their interface.

- The main proteins could have been identified by mass spectrometry.

Our Response and Revision: Thanks for the reviewer's instructive suggestion. d-SMG is mainly composed of 30-50% protein and 10-16% snail glycosaminoglycan. Indeed, we did not further identify the main proteins except for analyzing its molecular weight through SDS-PAGE. According to the reviewer's valuable advice, we have identified the main proteins of the natural biomaterial by mass spectrometry. Our new results are as follows:

Briefly, we analyzed the d-SMG proteins by liquid chromatography-mass spectrometry (LC-MS/MS), collected the MS data and searched the known protein database UniProtKB

(<https://www.uniprot.org/uniprotkb?query=Stylommatophora>). We identified 156 and 44 proteins in d-SMG from the snail *A. fulica* and *H. lucorum*, respectively. The complete data are available in Source Data and/or ProteomeXchange with identifier PXD036781 (Reviewer account details: Username: reviewer_pxd036781@ebi.ac.uk, Password: R83ttMLM). Limited to the length of manuscript, only the top 80% (based on protein abundance) results are provided in this reply.

The results showed that the main proteins of the *A. fulica* d-SMG include hemocyanin, achacin, actin, cytoplasmic, APH domain-containing protein, and much uncharacterized proteins. And that of *H. lucorum* d-SMG had similar result except for some low abundance proteins. The data were listed in Supplementary Tables 3–6.

In consideration of the limitations of SDS-PAGE electrophoresis method mentioned by the reviewer in Comment 2, we further performed mass spectrometry analysis on the main protein bands obtained by SDS-PAGE analysis (Supplementary Fig. 6c). For the *A. fulica* d-SMG, most of the proteins with larger than 130 kDa (fraction AF1) and 100 kDa (fraction AF2) were belong to the hemocyanin family. The fraction AF3 contained plenty of Achacin (62.09 %), and the fraction AF4 (~45 kDa) had transporter and actin. The remaining small molecule proteins (AF5, ~15 kDa) were mainly histone and protein kinases. The protein fractions (HL1~6) from *H. lucorum* d-SMG had similar protein profile to that of *A. fulica* d-SMG. In short, the identification results of these protein bands are consistent with those of the original snail mucus (d-SMG). Based on the above results, the functions of these proteins might be worth to further explore. We have added this study in the resubmitted manuscript in Result section.

Supplementary Fig. 6c SDS-PAGE image of *A. fulica* d-SMG and *H. lucorum* d-SMG

Supplementary Table 3. Identified proteins in the *A. fulica* d-SMG by LC-MS.

Protein IDs (Uniprot)	Protein	Organism	Unique peptides	Unique sequence coverage	Protein abundance
A0A3G2VHN3	Hemocyanin β	Cornu aspersum	4	0.9 %	41.68 %
A0A0B7BT89	Uncharacterized	Arion vulgaris	8	3.7 %	8.24 %
A0A3G2VHF1	Hemocyanin α N	Cornu aspersum	5	1.7 %	6.74 %
A0A3G2VFW4	Hemocyanin α N	Helix pomatia	2	0.9 %	6.14 %
A0A3G2VFAQ5	Hemocyanin α D	Cornu aspersum	2	0.4 %	4.44 %
A0A0B7A203	Uncharacterized	Arion vulgaris	1	15.3 %	4.10 %
P35903	Achacin	Lissachatina fulica	28	44.8 %	3.65 %
A0A3G2VHT2	Hemocyanin β	Helix pomatia	1	0.4 %	3.54 %
A0A0B7A0D2	Uncharacterized	Arion vulgaris	2	3.7 %	3.41 %
A0A0B7BCN0	Uncharacterized	Arion vulgaris	1	3.3 %	2.29 %
A0A0B7AFX7	Uncharacterized	Arion vulgaris	1	2.7 %	2.14 %
A0A0B7APA2	Uncharacterized	Arion vulgaris	1	2.9 %	2.06 %
A0A0B6ZQQ7	Haemocyan_bet_s domain-containing	Arion vulgaris	1	12.2 %	1.91 %
A0A0B7A255	Uncharacterized	Arion vulgaris	2	7.1 %	1.28 %
A0A345S6Z0	Actin, cytoplasmic	Cepaea nemoralis	13	38.8 %	1.18 %
A0A0B7AHS6	APH domain-containing	Arion vulgaris	1	3.3 %	1.00 %
A0A0B6YGU9	Uncharacterized	Arion vulgaris	1	28.2 %	0.98 %
A0A0B7B9A6	Uncharacterized	Arion vulgaris	1	3.4 %	0.67 %

Supplementary Table 4. Identified proteins in the *H. lucorum* d-SMG by LC-MS.

Protein IDs (Uniprot)	Protein	Organism	Unique peptides	Unique sequence coverage	Protein abundance
G3FPE6	Hemocyanin α D-subunit	Helix lucorum	26	7.9 %	37.14 %
G3FPE5	Hemocyanin β -subunit	Helix lucorum	46	18.4 %	28.18 %
G3FPE7	Hemocyanin α N-subunit	Helix lucorum	36	18.1 %	23.58 %
A0A3G2VFW4	Hemocyanin α N	Helix pomatia	16	5.9 %	3.43 %
A0A3G2VHT2	Hemocyanin β	Helix pomatia	18	8.0 %	2.96 %
A0A3G2VHR9	Hemocyanin α D	Helix pomatia	13	3.4 %	1.93 %
A0A3G2VHF1	Hemocyanin α N	Cornu aspersum	8	3.8 %	0.99 %
A0A3G2VFAQ5	Hemocyanin α D	Cornu aspersum	13	3.9 %	0.68 %
A0A0B7AMR8	Uncharacterized	Arion vulgaris	3	4.4 %	0.64 %
A0A3G2VHN3	Hemocyanin β	Cornu aspersum	15	5.6 %	0.16 %
A0A0B7B0J4	Uncharacterized	Arion vulgaris	2	4.0 %	0.09 %
A0A345S6Z0	Actin, cytoplasmic	Cepaea nemoralis	3	14.6 %	0.06 %
A0A0B6ZIIY0	PKS_ER domain-containing	Arion vulgaris	2	7.7 %	0.02 %
A0A345S6Y9	Tubulin α chain	Cepaea nemoralis	3	10.0 %	0.02 %
A0A0B7C4D3	Tyrosinase_Cu-bd domain- containing	Arion vulgaris	1	18.6 %	0.02 %
A0A0B7A345	Phosphoenolpyruvate carboxykinase (GTP)	Arion vulgaris	1	2.9 %	0.01 %

A0A0B7B670	Tubulin β chain	Arion vulgaris	1	4.0 %	0.01 %
P00994	Isoinhibitor K	Helix pomatia	1	24.1 %	0.01 %

Supplementary Table 5. Identified proteins in the SDS-PAGE of *A. fulica* d-SMG by LC-MS

SDS-PAGE Fractions	Protein ID (Uniprot)	Protein	Organism	Unique peptides	Unique sequence coverage	Protein abundance
AF1 (> 130 kDa)	A0A3G2VHN3	Hemocyanin β	Cornu aspersum	6	1.9 %	38.31 %
	A0A3G2VHF1	Hemocyanin α N	Cornu aspersum	5	1.7 %	9.56 %
	A0A0B7BT89	Uncharacterized	Arion vulgaris	5	2.2 %	8.44 %
	A0A3G2VHT2	Hemocyanin β	Helix pomatia	0	0.0 %	6.16 %
	A0A3G2VHR9	Hemocyanin α D	Helix pomatia	0	0.0 %	4.95 %
	A0A0B7AMR8	Uncharacterized	Arion vulgaris	3	5.4 %	4.53 %
	A0A3G2VFW4	Hemocyanin α N	Helix pomatia	4	1.5 %	4.24 %
	A0A0B7APA2	Uncharacterized	Arion vulgaris	1	2.9 %	3.06 %
AF2 (~100 kDa)	G3FPE6	Hemocyanin α D-subunit	Helix lucorum	1	0.2 %	33.25 %
	A0A3G2VHF1	Hemocyanin α N	Cornu aspersum	7	2.5 %	12.81 %
	A0A3G2VHN3	Hemocyanin β	Cornu aspersum	3	0.7 %	10.84 %
	A0A0B7AFX7	Uncharacterized	Arion vulgaris	1	2.7 %	8.84 %
	A0A0B7BT89	Uncharacterized	Arion vulgaris	5	2.5 %	8.24 %
	A0A3G2VFQ5	Hemocyanin α D	Cornu aspersum	2	0.4 %	7.53 %
AF3 (~60 kDa)	P35903	Achacin	Lissachatina fulica	27	43.3 %	62.09 %
	A0A3G2VHN3	Hemocyanin β	Cornu aspersum	2	0.3 %	6.42 %
	A0A3G2VFQ5	Hemocyanin α D	Cornu aspersum	2	0.4 %	5.72 %
	A0A0B7AFX7	Uncharacterized	Arion vulgaris	1	2.7 %	4.00 %
	A0A0B6ZW97	Uncharacterized	Arion vulgaris	1	27.4 %	2.84 %
AF4 (~45 kDa)	A0A0B7AZK1	Transporter	Arion vulgaris	1	1.0 %	81.28 %
	A0A345S6Z0	Actin, cytoplasmic	Cepaea nemoralis	0	0.0 %	8.56 %
	G3FPE6	Hemocyanin α D-subunit	Helix lucorum	0	0.0 %	1.71 %
	A0A3G2VFW4	Hemocyanin α N	Helix pomatia	1	0.4 %	1.14 %
	A0A0B7BT89	Uncharacterized	Arion vulgaris	2	0.5 %	0.923 %
	A0A220VX78	Actin	Gibbulinella dewinteri	1	3.8 %	0.612 %
	M1HH77	Histone H4	Chondrina avenacea	4	30.2 %	25.48 %
	A0A0B7A7U4	Protein kinase domain-containing	Arion vulgaris	1	1.5 %	11.80 %
	A0A0B7C5B8	Protein kinase domain-containing	Arion vulgaris	1	16.4 %	11.24 %
	A0A0B7AN25	Multiple inositol polyphosphate phosphatase 1	Arion vulgaris	1	1.6 %	7.56 %
AF5 (~15 kDa)	A0A0B7A1A5	Golgi apparatus membrane protein TVP23 homolog	Arion vulgaris	1	3.2 %	5.34 %
	A0A3G2VFQ5	Hemocyanin α D	Cornu aspersum	1	0.3 %	4.87 %
	A0A0B6XU90	XPGN domain-containing	Arion vulgaris	1	12.1 %	4.32 %

	A0A0B6Y9Y9	Histone H2B	Arion vulgaris	8	45.3 %	3.97 %
	A0A0B6ZNS7	J domain-containing	Arion vulgaris	1	2.0 %	3.85 %
	A0A0B6Y975	Histone H2A	Arion vulgaris	1	5.5 %	3.19 %
Supplementary Table 6. Identified proteins in the SDS-PAGE of H. lucorum d-SMG by LC-MS						
SDS-PAGE Fractions	Protein ID (Uniprot)	Protein	Organism	Unique peptides	Unique sequence coverage	Protein abundance
HL1 (>130 kDa)	G3FPE6	Hemocyanin α D-subunit	Helix lucorum	34	7.8 %	36.54 %
	G3FPE5	Hemocyanin β -subunit	Helix lucorum	54	20.2 %	27.00 %
	G3FPE7	Hemocyanin α N-subunit	Helix lucorum	40	20.2 %	25.01 %
	A0A3G2VFW4	Hemocyanin α N	Helix pomatia	21	7.9 %	3.33 %
	A0A3G2VHR9	Hemocyanin α D	Helix pomatia	17	5.0 %	2.63 %
HL2 (~60 kDa)	G3FPE6	Hemocyanin α D-subunit	Helix lucorum	6	1.7 %	18.02 %
	G3FPE7	Hemocyanin α N-subunit	Helix lucorum	8	4.7 %	15.04 %
	A0A0B7B016	N domain-containing	Arion vulgaris	1	7.0 %	14.28 %
	G3FPE5	Hemocyanin β -subunit	Helix lucorum	6	1.9 %	13.51 %
	A0A0B6ZSP8	Uncharacterized	Arion vulgaris	1	2.7 %	12.77 %
	A0A0B6YKV0	Uncharacterized	Arion vulgaris	1	15.1 %	11.30 %
	A0A0B7A4J6	Katanin p60 ATPase-containing subunit A1	Arion vulgaris	1	2.8 %	3.82 %
	A0A3G2VFW4	Hemocyanin α N	Helix pomatia	4	1.7 %	1.74 %
HL3 (~40 kDa)	G3FPE6	Hemocyanin α D-subunit	Helix lucorum	7	2.2 %	16.04 %
	A0A0B6ZVB5	Protein kinase domain-containing	Arion vulgaris	1	1.5 %	14.64 %
	G3FPE5	Hemocyanin β -subunit	Helix lucorum	8	2.9 %	12.60 %
	G3FPE7	Hemocyanin α N-subunit	Helix lucorum	6	2.5 %	10.10 %
	A0A0B6Y997	Uncharacterized	Arion vulgaris	1	6.1 %	6.56 %
	A0A0B7A4J6	Katanin p60 ATPase-containing subunit A1	Arion vulgaris	1	2.8 %	4.91 %
	A0A0B6YX79	Uncharacterized	Arion vulgaris	1	17.6 %	4.40 %
	A0A0B6YH37	Uncharacterized	Arion vulgaris	1	4.2 %	4.19 %
	A0A0B6ZYW6	Uncharacterized	Arion vulgaris	1	23.3 %	3.26 %
	A0A0B6ZDR2	Myosin_tail_1 domain-containing	Arion vulgaris	1	2.5 %	3.19 %
	A0A0B7B0A4	AB hydrolase-1 domain-containing	Arion vulgaris	1	2.8 %	2.67 %
	HL4 (~30 kDa)	A0A0B7BCQ8	Importin N-terminal domain-containing	Arion vulgaris	1	0.9 %
A0A0B7A456		Uncharacterized	Arion vulgaris	1	7.5 %	15.27 %
A0A0B6ZC31		DUF4200 domain-containing	Arion vulgaris	1	1.9 %	11.58 %
A0A0B6ZKG6		Death domain-containing	Arion vulgaris	1	6.2 %	8.15 %
A0A0B7AH50		Uncharacterized	Arion vulgaris	1	19.6 %	6.50 %
A0A0B7A4J6		Katanin p60 ATPase-containing	Arion vulgaris	1	2.8 %	5.22 %

		subunit A1				
	G3FPE6	Hemocyanin α D-subunit	Helix lucorum	6	1.5 %	5.11 %
	A6YM38	Matrilin-like 85 kDa	Ambigolimax valentianus	1	1.0 %	39.04 %
	A0A0B7BCQ8	Importin N-terminal domain-containing	Arion vulgaris	1	0.9 %	19.15 %
HL5 (~25 kDa)	A0A0B6ZY91	Uncharacterized	Arion vulgaris	1	16.0 %	16.63 %
	A0A0B7C6J4	Uncharacterized	Arion vulgaris	1	15.6 %	5.93 %
	A0A0B7A456	Uncharacterized	Arion vulgaris	1	7.5 %	4.56 %
	A0A0B6Y1H3	Uncharacterized	Arion vulgaris	1	6.4 %	3.24 %
	A0A0B7A4J6	Katanin p60 ATPase-containing subunit A1	Arion vulgaris	1	2.8 %	1.93 %
	A0A0B7AQC8	Uncharacterized	Arion vulgaris	1	0.9 %	9.93 %
	G3FPE6	Hemocyanin α D-subunit	Helix lucorum	8	2.8 %	9.89 %
	A0A0B6YNX5	Uncharacterized	Arion vulgaris	1	8.2 %	9.22 %
	G3FPE5	Hemocyanin β -subunit	Helix lucorum	4	1.5 %	8.01 %
	G3FPE7	Hemocyanin α N-subunit	Helix lucorum	4	1.5 %	6.98 %
	P00994	Isoinhibitor K	Helix pomatia	1	24.1 %	5.91 %
	A0A1X9WEF7	Fmrfamide-related peptide 2	Deroceras reticulatum	1	4.3 %	5.86 %
HL6 (~15 kDa)	A0A0B6Z4T8	Uncharacterized	Arion vulgaris	1	13.5 %	5.78 %
	A0A0B6Z5Q5	SynN domain-containing	Arion vulgaris	1	16.7 %	4.77 %
	A0A3G2VHR9	Hemocyanin α D	Helix pomatia	2	0.4 %	3.99 %
	A0A0B7AQB0	3-phosphoinositide-dependent protein kinase 1	Arion vulgaris	1	1.4 %	3.36 %
	A0A0B7A8X1	Uncharacterized	Arion vulgaris	1	3.2 %	3.06 %
	A0A0B6ZFN8	Uncharacterized	Arion vulgaris	1	5.1 %	3.02 %
	A0A0B7BL84	Uncharacterized	Arion vulgaris	1	2.4 %	2.71 %
	M1HH77	Histone H4	Chondrina avenacea	5	50.0 %	2.59 %

2. The authors provide a molecular weight based on the SDS-PAGE analysis of the soluble fractions of both secretions. This seems to have been the only analysis done in this way. It is unclear what proportion of the material is soluble or insoluble. What is the role of the insoluble fraction?

Our Response and Revision: Thank the reviewer for these good suggestions. Further protein analysis of soluble fractions based on SDS-PAGE has been described in Comment 1. This is an interesting question about the insoluble substances that may exist in this material. During the preparation of SDS-PAGE samples, no obvious insoluble substance was observed in original snail mucus. Since d-SMG is a gel-like substance, we diluted the newly collected SMG by 25 times to obtain some insoluble substances through high-speed centrifugation ($15000 \times g$). We did not observe any obvious insoluble substance.

3. Could the authors comment on the use of fibrin as an adhesion comparison group. How would other approved tissue glues perform compared to the d-SMG? I see that the authors have included a cyanoacrylate comparison for

Figure 5 which could be interesting to look at in terms of adhesion.

Our Response and Revision: Thanks for the reviewer's kindly suggestion. We have added some objective comments on the use of fibrin as an adhesion comparison group. Fibrin glues have been widely used as a hemostatic agent and tissue sealant in surgical applications since it was approved by FDA in 1998. (Duarte, A. P. et al. *Prog. Polym. Sci.* 37, 1031–1050, 2012; Seyednejad, H. et al. *Br. J. Surg.* 95, 1197–1225, 2008). However, fibrin glues also have some limitations, for instance, the risk of transmission of blood-borne disease, and the relatively poor mechanical properties with low cohesion strength (Nam, S. et al. *Chem. Rev.* 121, 11336-11384, 2021). Given that d-SMG is also a bioadhesive, and that commercial fibrin glue is commonly used as control in adhesion strength tests in literatures, we selected fibrin glue as the main control in this study to evaluate the biomaterial property of d-SMG. Additionally, it's reported that fibrin glues require a relatively long time to form adhesion (longer than 1 min) (Yuk, H. et al. *Nature* 575, 169-174, 2019). While d-SMG can form effective adhesion within 15 seconds after contacting wet tissue.

As for other commercial tissue adhesives include cyanoacrylate (CA), it's a well-known synthetic adhesive, with very potent adhesion strength. Based on the suggestions of reviewer, we used the same evaluation system in the manuscript to evaluate the shear adhesion strength of CA (shown below); as expected it far outperformed the biologically derived adhesives (about 10 folds of fibrin glue and 5.8~8.8 folds of d-SMG). However, CA displayed poor biodegradability (Figure 5), and had potential toxicity due to the exothermic polymerization reaction and residue (Bal-Ozturk, A. et al. *Nano Today* 36, 101049, 2021).

Supplementary Fig. 8c Adhesion strength of d-SMGs, fibrin glue and cyanoacrylate

Data are presented as the mean \pm SEM, $n = 3$, $**P < 0.01$, two-tailed t-test.

4. Figure 1. The SEM image were of freeze-dried samples. This method of dehydration is known to form large pores into the materials due to the formation of ice crystals. The representatives SEM images thus do not bring a lot of information.

Our Response and Revision: We agree with the reviewer's point. Indeed, the dehydration process may produce larger or additional pores, which would interfere the analysis of SMG microstructure. Therefore, we have removed the SEM image from Figure 1 and revised the corresponding results in the new manuscript to avoid misleading readers.

5. Figure 1j, k. Could the authors write in the figure legend that the percentages are hydration%. I had to look it up

elsewhere, Also, it would be good to understand what the % hydration of the fresh samples is in comparison (relates back to my previous question on d-SMG vs fresh-SMG).

Our Response and Revision: We have added this information in the figure legend. Thanks for the reviewer's suggestion.

6. I am sure the authors say it somewhere, it is not clear what hydration was used for all the evaluations passed the rheology. Please indicate that in the text, and ideally in the name of the d-SMG in all subsequent figures.

Our Revision: We did mention it in the Method section (**Characterization**). In the resubmitted manuscript, we have also added the information of hydration% in the text and figures.

7. Page 4 – line 143. It is not clear why the authors think that the d-SMGs are stronger than their fresh counterparts. Is that for an equivalent hydration rate?

Our Response and Revision: We have deleted this inaccurate statement. Base on the average yield of d-SMG (3.65% from *A. fulica* and 5.26% from *H. lucorum*), the hydration rate (> 90%) of fresh SMG was higher than that of the d-SMG (when used in assays it's dissolved in water with the hydration rate of 66.7%). We are sorry for not considering the hydration rate when compared their adhesive strengths.

8. Page 4 line 153 – The authors mention 5% hemolysis as a threshold for biocompatibility. Please add a reference to back that up.

Our Revision: We have added the following reference in the resubmitted manuscript.

26. Yao, M., Wei, Z., Li, J. et al. Microgel reinforced zwitterionic hydrogel coating for blood-contacting biomedical devices. *Nat. Commun.* **13**, 5339 (2022).

9. Figure 3. Regarding the cytocompatibility data. The s-SMG show increase in metabolic activity that seems to go beyond experimental variations. This could be linked with increase proliferation or metabolic activity. Could the authors look into this and discuss?

Our Response and Revision: In consideration of the reviewer's comments, we also noticed that previous data indicated that d-SMG (0.2 mg/ml) might promote fibroblasts (L929 cell lines) proliferation at 48 h, which could benefit wound healing.

Given that in previous results the effects of d-SMGs after incubation for 48 h had a large discrepancy, to make sure whether the hypotheses worth further study, we first repeated the cell viability assays using both MTT test and real-time recording of cell growth by IncuCyte S3 Live-Cell Analysis System (Sartorius, Germany). The MTT test and real-time recording were conducted in 10 and 5 repeats, respectively. The results from both assays showed that d-SMGs have no significant effects on cell viability of fibroblasts (L929 cell lines) within 72 h (Figures shown below). We provided these new data in the resubmitted manuscript (Figure 3a and Supplementary Figure 9a).

Effect of d-SMG on cell viability and cell growth

a The cell viability of L929 was detected by MTT method ($n = 10$). **b** The real-time cell growth of L929 ($n = 5$). Data are presented as the mean \pm SEM.

10. Please explain what concentration is used for the cytocompatibility assay. It is not mentioned on page 20 in the methods either. That seems like an important information.

Our Revision: We are sorry for missing the important information. The concentration of d-SMG is 0.2 mg/mL in the cytocompatibility assay. This information has been added in both the figure legend and Method section.

11. Figure 6. Regarding the effect of d-SMG on wound healing. I might have missed it, but it is not clear from the manuscript what is the GAG that was used as a control in the wound healing studies (Figures 6 and 7). This is especially important to know given the good results the GAG conditions showed in these studies compared to the d-SMG

Our Response and Revision: We are sorry for the incaution, and thanks for the reviewer's reminding. Generally, GAG refer to glycosaminoglycan such as heparin, sulfated heparan, chondroitin sulfate, et al. The GAG group mentioned herein refers to the snail GAG extracted from the *A. fulica* d-SMG. We have replaced GAG with s-GAG (snail glycosaminoglycan) to distinguish it from other GAGs in the resubmitted manuscript.

12. It is a bit surprising to me to see that although the diabetic mice model does provide slower wound closure time than the healthy mice, wound closure in untreated animals is still relatively rapid. Would this be considered chronic wound? Could the authors comments on the limits of this model in the context of this material testing?

Our Response and Revision: This is a very good recommendation. We thank for the reviewer's careful reading and good suggestion.

Chronic wounds refer to the barrier defects that do not progress through orderly and timely reparation to recover the structural and functional integrity, characterized with persistent inflammation (Eming, SA. et al. *Sci. Transl. Med.* 6, 265sr6, 2014). For the normal wound repair in mammals, the inflammation stage lasts until about 48 h after injury, and the new tissue formation stage occurs about 2~10 days after injury (Gurtner, GC. et al. *Nature* 453, 314-21, 2008). According to the current knowledge in this field, the wound model in STZ-induced diabetic rats should be considered as chronic wound, and it's commonly used to study chronic DFU (Bardill, J. R. *Acta Biomater.* 138, 73-91, 2022).

Although consensus is lacking regarding the animal model that best recapitulate human DFU pathophysiology, streptozotocin (STZ)-induced diabetic models are one of the most studied models of diabetic wound healing. In

practice, the wound healing process in STZ-induced rats depends on the STZ dose, animal age, diet, wound size and number etc., ranging from 14 to 21 days. (Kaur, P. et.al. *Nanomed. Nanotechnol. Biol. Med.* 15, 47–57, 2019; Xu, Q. et.al. *Chem. Sci.* 9, 2179–2187, 2018). Our data on the untreated group was consistent with that reported by others using the similar wound model (Valizadeh, A. et.al. *Drug Deliv. Transl. Res.* 11, 292–304, 2021; Kaisang, L. et.al. *J. Surg. Res.* 217, 63–74, 2017).

The wound model in STZ-induced diabetic rats is recognized as chronic wound. However, this model has limits in mimicking human DUF in terms of pathophysiology, for instance it may not have the similar peripheral neuropathy, prolonged systemic hyperglycemia and mechanical stress etc. Nevertheless, our data have exhibited optimistic results for the concept of using natural adhesive in wound healing. Further assessment of d-SMG application may require animal models with delayed wound healing such as db/db mice and porcine, which are more closely resembles the physiology of human skin to illustrate the translational and clinical potential (Hackl, F. et al. *Adv. Wound Care* 2, 166-170, 2011; Eming, S. A. et al. *Sci. Transl. Med.* 6, 265sr6, 2014).

We have added these objective comments on the limits of this model in Discussion in our new manuscript.

13. It seems like the angiogenesis markers are indeed upregulated in the diabetic wound model for d-SMG and GAGs. But since the authors have nice immunohistochemistry done on these tissues, would it not also make sense to directly measure the number of vessels and their size as a more direct proof of angiogenesis?

Our Response and Revision: Thanks for the reviewer’s great proposal. We have analyzed the number and diameter of vessels based the immunofluorescence images. The data showed that both d-SMG and s-GAG significantly increased the number of vessels in wound tissue on day 7. We have provided this information in the resubmitted manuscript in Figure 6 and Results section.

Fig. 6 Effects of d-SMG on wound healing in a diabetic wound model

.....h Representative images of CD31 and α -SMA immunostaining in wound samples on day 7 and 14. i-j Quantification of the number (i) and diameter (j) of vessels in the wound tissue ($n = 9$). Data are presented as the mean \pm SEM, $*P < 0.05$, two-tailed t-test.

14. Figure 7 Regarding inflammatory regulation. The effect of GAG and Alginate is strong in the immune regulation. The authors have not commented or discussed this result in depth. How does it compare to the literature? It was not my impression that alginates could have such a strong immune dampening effect.

Our Response and Revision: We thank for the reviewer’s kindly reminding. The results in Fig. 7 showed effects of d-SMG, s-GAG and Alginate on macrophage regulation and inflammation in the wound tissues in diabetic rats.

On day 3, the high expression of inflammatory cytokines in control group (without treatment) reflected the inflammation phase in the diabetic chronic wound. In this case, both s-GAG and alginate were able to reduce the level of inflammatory cytokines in wound tissue (Fig. 7d). We further showed that s-GAG may also have direct

effect of promoting the polarization of macrophage (RAW264.7) to M2 phenotype (revised Fig. 7e-f). The mechanism of macrophage polarization by s-GAG was further explored (referred to Comment 16).

As for Alginate dressing, it's reported that it can efficiently absorb wound exudate *in vivo*, and the moist dressing should be replaced every 2-3 days (Williams, C. *Br. J. Nuts.* 7, 550-2, 1998). According to the instruction of this biomaterial, the alginate dressing can absorb exudate as much as up to 20 times their own weight (<https://multimedia.3m.com/mws/media/432453O/tegaderm-alginate-dressing-product-application-guide.pdf>).

During our experiment, we also observed the absorption effect of Alginate. When we collected the wound tissue sample to determine the inflammatory cytokine levels, the Alginate dressing was not included. The reduced inflammation cytokine levels observed in AlgTM group may be due to the removal of used dressings together with the absorbed tissue exudate.

15. The authors have not shown the effect of Alginate and GAG on the macrophage cell line Figure 7f. This would be interesting to support the *in vivo* results.

Our Response and Revision: This is a very good suggestion. We have carried out the additional tests to investigate effects of s-GAG and AlgTM on the macrophage cell line. The s-GAG significantly promoted the polarization of macrophages to the M2 phenotype (CD206⁺), more than 2 folds of the negative control, slightly stronger than d-SMG and weaker than the positive control IL-4, whereas AlgTM had no obvious effect. We have added these data in resubmitted manuscript (Figure 7e and 7f).

Fig. 7 Effects of d-SMG for inflammatory regulation

e Representative images of CD206 and DAPI staining of RAW264.7. **f** Relative fluorescence intensity of CD206 in RAW364.7 after treatments ($n = 10$). For **f**, data were expressed as mean \pm SEM, *** $P < 0.001$, two-tailed t-test.

16. The authors mention that the GAG fraction in the d-SMG could capture inflammatory cytokines and participate in the immune modulatory effect. Could the authors discuss more their data in that context? Do the authors think that the low cytokines levels measured are due to their sequestering, which prevent them from being sampled? Or are their biological effects leading to the inhibition of their secretions? How does the *in vitro* assay on macrophage inform you in this regard?

Our Response and Revision: Thanks for the reviewer's instructive comments. In our original manuscript, the statement that s-GAG can capture inflammatory cytokines mentioned is based on the structural analysis of s-GAG and literatures (Lohmann, N. et al. *Sci. Transl. Med.* 9, eaai9044, 2017). Benefited from the suggestions of reviewer, we further determined the interactions between s-GAG and inflammatory cytokines by surface plasmon resonance (SPR) assays. We found that s-GAG can bind to inflammatory cytokines, including TNF- α , IL6, IL8, and IP10, with

high affinities (K_D , 0.01–0.34 μM), which was similar to that of heparin (K_D , 0.17–0.61 μM) (Supplementary Figure 16). The results are consistent with the report that negative-charged glycosaminoglycan can efficiently bind to the cytokines with positive-charged amino acids (Handel, TM. et al. *Annu Rev Biochem.* 74, 385–410, 2005; Lohmann, N. et al. *Sci. Transl. Med.* 9, eaai9044, 2017). This suggests that s-GAG in the d-SMG can capture inflammatory cytokines, thus reducing the inflammatory cytokines in wound site.

Additionally, in the chronic diabetic wound both d-SMG and s-GAG promoted the polarization of macrophages toward M2 phenotype (Fig. 7). To further explore the mechanism, we analyzed the effects of d-SMG and s-GAG on the signaling protein and mRNA levels in macrophages, by Western Blot and qRT-PCR, respectively. Compared with the control, d-SMG and s-GAG increased the phosphorylation level of signal transducer and activator of transcription 3 (STAT-3), whereas they had no obvious effect on the phosphorylation of STAT1 and STAT6 (Supplementary Fig. 15). These results indicated the important role of STAT3 in their function in promoting M2 polarization of macrophages (Sica, A. & Mantovani, A. *J. Clin. Invest.* 122, 787–795, 2012). Moreover, they also significantly upregulated the mRNA expression levels of *Arg-1* and *Il-10*, which are associated with macrophage M2 phenotype (Supplementary Fig. 15).

Taken together, these data suggested d-SMG and s-GAG may attenuate inflammation in diabetic wound by capturing inflammatory cytokines for their binding ability, and by reducing secretion of inflammatory cytokines via promoting the polarization of macrophages toward the anti-inflammatory M2 phenotype. We have added these results and discussion in the resubmitted manuscript.

Supplementary Fig. 16. Binding ability of s-GAG to inflammatory cytokines. a–b The association and dissociation dynamic curves of s-GAG (a) and heparin (b) binding to immobilized cytokines. c The affinity of s-GAG and heparin to cytokines.

Supplementary Fig. 15. Effects of d-SMG and s-GAG for inflammatory regulation *in vitro*. a–c The protein levels of p-STAT-1 (Y701), p-STAT-3 (Y705), and p-STAT-6 (Y641) in RAW264.7 treated by d-SMG (0.2 mg/ml) and s-GAG (0.15 mg/ml), IFN- γ (10 ng/ml), IL-10 (20 ng/ml), and IL-4 (5 ng/ml), assessed by western blot. d–f Relative density of p-STAT-1/STAT-1 ($n = 3$), p-STAT-3/STAT-3 ($n = 5$), and p-STAT-6/STAT-6 ($n = 3$) from western blot images. g–i Relative mRNA level in macrophages assessed using real-time PCR. For d–i, data were expressed as mean \pm SEM, * $P < 0.05$, ** $P < 0.01$, and *** $P < 0.001$ vs. control, two-tailed t-test.

17. Could the authors also enrich the discussion by describing what they see for this material. Can scaleup of production be achieved? In that case, can sterilization be achieved without affecting the properties? Would the next step be to further dissect the material composition and create synthetic versions?

Our Response and Revision: Thanks to the reviewer for the good suggestion, which greatly enrich the interpretation and discussion of our manuscript. Our data on its biomaterial properties and *in vivo* and *in vitro* activities support that the d-SMG may be a potential wound dressing as a natural degradable bioadhesive. Its adhesion performance is stronger than the commercial fibrin glue. Its natural gelation mechanism also has high value for biological enlightenment, which provides some basic theories for the development of a new generation of synthetic tissue adhesives.

Snails have been recognized for centuries for human consumption, obtained either by aquaculture or capture (Paszkiwicz, Kozyra & Rzeżutka, *Food Anal. Method.* 8(3), 799–806, 2014). The snail *A. fulica* constitutes the predominant part of land snails in Chinese food markets (Lv, Zhang, Steinmann & Zhou, *Emerg. Infect. Dis.* 14, 161–164, 2008). In Taiwan of China, this species is the main material of a dish called hot frying snails, which is one of the traditional snacks, both delicious and nutritious. A common variety of the snail *A. fulica*, which is called China white jade snail, has been successfully cultured (Vieira et al., *Eur. J. Biochem.* 271, 845–854, 2004). The

annual output in China can reach thousands of tons (“World: Snails (Except Sea Snails) -Market Report, Analysis and Forecast to 2025”, 2017). The preparation process of d-SMG is not difficult, and the scaleup production is feasible. Presently, we developed the whole standardized preparation process (also see our Response for question 4 from Reviewer #3). We also compared the main components (protein, s-GAG, and Na) of d-SMG from random 3 batches, which showed very small discrepancy (Supplementary Table 9).

Standard preparation procedure of d-SMG

Minor issues:

Figure 4e. That graph has a different style than others, it would be good to harmonize

Our Revision: We have modified Figure 4e in the resubmitted manuscript.

Page 4 line 179 – “The control group had...” Please just say what that group was. Is it fibrin?

Our Revision: The control group means no other hemostatic treatment; which is a negative control. We have revised “The control group...” to be “The negative control group...” in the resubmitted manuscript.

Page 3, line 123. Reference 21 seems a bit misplaced here. Is heparin really secreted in intestinal mucus or is it part of the tissue

Our Response and Revision: We have changed Reference 21 “Liu, H., Zhang, Z. & Linhardt, R. J. Lessons learned from the contamination of heparin. *Nat. Prod.* 711 Rep. 26, 313–321 (2009).” to be “Linhardt, R. J. 2003 Claude S. Hudson Award Address in Carbohydrate Chemistry. Heparin: Structure and Activity. *J. Med. Chem.* 46, 2551–2564, (2003).” Heparin is from part of the tissue, in our original manuscript, the example of heparin source was “mucosa”, not “mucus”.

The commercial heparin is mostly extracted from beef lung and porcine mucosa. Sorry for the unclear comment in line 123. The sentence “Notably, the specific secretion of heparin-like GAG in invertebrates is rare; it has been reported mostly in mammals, such as porcine small intestinal mucosa²¹” was revised to be “Notably, compared with source of invertebrate body and mammal tissue^{19–21}, the abundant heparin-like GAG in invertebrate mucus is rarely reported.”

Page 6, section on Inflammatory regulation line 267. It is unclear why IL-1Beta is mentioned before and separately than the other inflammatory markers mentioned in the next phrases.

Our Response and Revision: Since IL-1 β is the main inflammatory cytokine secreted from M1 macrophage, at first, we tested its level in the wound tissue by *in situ* immunofluorescence. Then to evaluate the overall effect of treatment on inflammation, we tested the levels of multiple inflammatory cytokines collected from wound tissue by

bio-plex assay.

Figures 5c, 6c, 6g indicate “rates” on the y-axis. These values are not technically rates since they are provided for a given time, and not relative to time.

Our Revision: We have replaced “rate” with “ratio” in the resubmitted manuscript.

Figure 7a. The conditions are not presented in the same order as on other graphs. Please keep the order the same.

Our Revision: We have modified the order in Figure 7a, to keep the same order as other graphs.

Figure 8c, 8d. It is not clear what the numbers on the scale are.

Our Revision: The numbers on the scale represent the gene expression level, which is normalized to that of FPKM (Fragments Per Kilobase of exon model per Million mapped fragments). We have added this information in the figure legend.

Figure 8e, 8f, what in “cound”. Is “rich factor” a typo. Did you mean “enrichment factor”

Our Revision: We have corrected the spelling mistake, “cound” should be “count”. “Rich factor” refers to the proportion of down-regulation gene in the total genes of a biological process based on the Gene Ontology. “Enrichment factor” should be more accurate and in common use. We have changed “rich factor” to be “enrichment factor”.

Page 21, line 599. D-SMG instead of SMG. Please indicate the controls used in these experiments.

Our Revision: We have replaced “SMG” with “d-SMG”, and added the information on controls in the Method section.

Reviewer #2 (Remarks to the Author):

REFEREE'S RECOMMENDATIONS

Accept with major changes.

Comments for the Authors:

The study contains clear and promising data and well written. However, I have some suggestions as below.

Our Response: Thank [redacted due to referee wishes] for very valuable and helpful comments for revising and improving our paper.

Line 218 & line 219: Authors must add *A. fulica* before d-SMG group in both lines. Also, the same comment on caption of figure 11 in supplementary data.

Our Response and Revision: Thanks for the good suggestions. To make it clear, we have added *A. fulica* before d-SMG, and other places where needed in the resubmitted manuscript.

Line 302: Authors must add magnification power of scanning microscope.

Our Revision: We have added the magnification power of scanning microscope in the new text.

Line 305: Revise it. I think it may be for I not for c.

Our Revision: Sorry for the writing mistake, we have corrected it.

Page 10 Fig 3d: Word day must be capitalized.

Our Revision: We have changed “d” to be “D”.

Page 14 Fig 7: Why authors didn't put results of day 14 as mentioned in material??? It must be clarified.

Our Revision: The immunofluorescence staining analysis of CD206 and CD86 was performed on day 3, 7 and 14. Because we mainly focused on the phenotypic change of macrophages on day 3 and 7 (the early stage of wound healing), the images on day 14 was not shown in Figure 7 in our old manuscript. These results have been added in the Supplementary Figure 13 and mentioned in the new text.

Page 14 Fig 7 caption: Which day did you mean???? Day 3 or 7 or 14. Revise it

Our Revision: Sorry for the mistake, we have revised “day 7 and 14” to be “day 3 and 7”

Line 492: *Achatina fulica* and *Helix lucorum* must be italic.

Our Revision: We have corrected the font of these two species names.

Line 508: Why authors write fresh SMG or rehydrated??? Which one do you use exactly??? It must be clarified.

Our Revision: Thanks for the reviewer's carefulness. It's a spelling mistake. We have changed “or” to be “and”. We wanted to describe that several hydrogels were prepared and tested for rheological property.

Page 19: Authors must add separation and purification method of GAG from d- SMG after characterization or in supplementary material after chemical composition analysis.

Our Revision: Thanks for the reviewer's reminding. We have provided the information on extraction and purification method of s-GAG in supplementary text (Method 2.4).

Line 515: Authors must write and instead of or.

Our Revision: Thanks for the reviewer's suggestions. We have changed “or” to be “and”.

Line 533: Authors must mention effect of GAG of 2 snails on human coagulation plasma in material as in results. It must be clarified.

Our Response and Revision: We have checked, and we did have this information in the results and Fig. 3c. The effects of GAGs, as well as d-SMGs on human coagulation plasma were evaluated by the activated partial thromboplastin time (APTT) assays. The results were mentioned in the text (the section of Biocompatibility and biodegradability): “The tests of activated partial thromboplastin time (APTT) indicated that neither d-SMGs nor their GAGs exhibited APTT-prolonging activity (Fig. 3c), suggesting that they had no bleeding risk.”

Line 551: ad libitum must be italic.

Our Revision: We have corrected the font to be italic.

Line 575: Authors must write name of apparatus and its company of scanning microscope.

Our Revision: We have provided this information in the new text.

Line 577& 578: in vivo, *Achatina fulica* and *Helix lucorum* must be italic. Please revise.

Our Revision: We have revised the font of these words to be italic.

Line 585& 591: in vivo and ad libitum must be italic.

Our Revision: We have revised it to be italic.

Line 599& 610: Revise, authors use SMG or D- SMG as in results.

Our Revision: We have revised SMG to be d-SMG in these two places.

Page22 line 604: In this section, why authors didn't use GAG in wound healing in normal rats, also??? Why in diabetic only???

Our Response and Revision: Thanks for the reviewer's valuable advice. In order to make our work more consistent, we also supplemented the assays testing the effect of s-GAG and AlgTM on wound healing in normal rats. The s-GAG group showed the best healing effect, and the wound healing ratio reached $92.7 \pm 1.11\%$ on day 11, which was significantly higher than that of the control group ($83.2 \pm 1.28\%$) (Supplementary Fig. 11). Different from the wound model of diabetes rats, the AlgTM dressing did not show a tendency to promote wound healing in normal rats. H&E staining showed that the *A. fulica* d-SMG and s-GAG group had thicker granulation tissue on day 5 (Supplementary Fig. 12). Masson's trichrome staining showed more collagen deposition in the granulation tissue

of the *A. fulica* d-SMG group on day 15 than in the control group (Supplementary Fig. 12). Furthermore, immunofluorescence staining analysis (CD31 and α -SMA) showed more neovascularization in the wound area in the *A. fulica* d-SMG group ($39.7 \pm 3.3/\text{mm}^2$) and *A. fulica* s-GAG ($26 \pm 1.9/\text{mm}^2$) group than that in the control group ($17.2 \pm 1.1/\text{mm}^2$) (Supplementary Fig. 12). These *in vivo* studies indicated that *A. fulica* d-SMG and s-GAG effectively promoted wound closure. We have added these new results in the resubmitted manuscript.

Supplementary Figure 11. Effect of d-SMG and s-GAG on the wound healing in normal rats. **a** Schematic illustration of the wound model in normal SD rats. **b** Representative images of the wound area in response. **c** Schematic diagram of the dynamic wound healing process. **d** The wound healing ratio after treatments ($n = 10-14$). **e** H&E-stained images of the wound tissue on day 5 and 15; Scale bar, 1 mm, 200 μm . **f** Masson staining of the wound tissue for collagen deposition on day 5 and 15. Scale bar, 1 mm, 200 μm . **g-i** Quantification of the granulation tissue thickness (**g**), new epidermis thickness (**h**) and collagen deposition ratio (**i**) in wound tissue on day 5 and 15. For **g**, **h**, and **i**, data were expressed as mean \pm SEM, $n = 12$, $*P < 0.05$, $**P < 0.01$, and $***P < 0.001$, two-tailed t-test.

Supplementary Figure 12. Effects of d-SMG and s-GAG on angiogenesis in normal rats. **a** Representative image of CD31 and α -SMA immunostaining in wound tissue after treatments. **b** Quantification of the number of vessels in the wound tissue. **c** Quantification of the diameter of vessels in the wound tissue. For **b** and **c**, data were expressed as mean \pm SEM, $n = 12$, $***P < 0.001$, two-tailed t-test.

Reviewer #3 (Remarks to the Author):

This manuscript reported the efficacy of snail mucus to act as a bio-adhesive material, to be used in wound repair.

The results are quite novel. There are already papers on the wound repair activity of snail mucus. The novelty of this research is mainly associated with the creation of an adhesive matrix.

The need for innovative products to repair epithelial damages makes this research of interest, even if already several data are present in the literature (The Effectiveness of Snail Slime and Chitosan in Wound Healing. Agnes Sri Harti et al. International Journal of Pharma Medicine and Biological Sciences Vol. 5, No. 1, January 2016; Application of snail mucin dispersed in detarium gum gel in wound healing. Michael Adikwu et al. Scientific Research and Essays 2(6):195-198, 2007; The Protective Effect of Snail Secretion Filtrate in an Experimental Model of Excisional Wounds in Mice. Enrico Gugliandolo et al. Vet Sci. 2021 Aug 20;8(8):167.; HelixComplex snail mucus exhibits pro-survival, proliferative and pro-migration effects on mammalian fibroblasts. Trapella Claudio et al. Sci Rep. 2018 Dec 5;8(1):17665.).

The research is well performed but some concerns are present:

1) The preparation of snail mucus is not clearly stated. It is important to clarify if any filtration is performed.

Our Response and Revision: We have revised the preparation part to make it clearer. “.....By stimulating the snail foot, the fresh snail-mucus gel (SMG) was collected from the secretion of snails. Approximately 20 g of fresh SMG was collected from 10 adult snails each time, and the next collection was performed after breeding two weeks. The dried SMG (d-SMG) was obtained from SMG by lyophilization (SMG was placed in liquid nitrogen for 15 min, and then placed in a lyophilizer at a pressure lower than 75 Pa for 48 h).” In the preparation process, no filtration was performed.

2) No data on microbiological content of snail mucus is reported. As it is used on injured skin, it is necessary to prove the sterility of the product.

Our Response and Revision: For d-SMG used *in vivo*, irradiation sterilization was performed and stored in -20 °C under nitrogen atmosphere until use. We conducted sterility test on 10 batches d-SMG samples according to the relevant standards of the Chinese Pharmacopoeia (1101 Sterility Tests), and no bacteria were found in all the products. Additionally, during the animal experiments, we did not find any phenomenon similar to bacterial infection in the wound sites.

3) It is mandatory to present a table indicating the content of the snail mucus. It is important to comment on the possible differences between *Achatina* and *Helix lucorum* mucus.

Our Response and Revision: This is a good suggestion. We summarized the protein, s-GAG, metal elements, anions, and allantoin of the two d-SMGs in Supplementary Table 7. The protein content in *H. lucorum* d-SMG is higher, which may cause the adhesion strength of *H. lucorum* d-SMG to be higher than that of *A. fulica*. The *A. fulica* d-SMG contains more s-GAG, which may be related to its effect of promoting wound healing.

Table 7. The content of components in two d-SMGs

Component	A. fulica	H. lucorum
Protein (%)	34.1±1.71	46.0±0.47
s-GAG (%)	15.9±0.11	9.3±0.11
Allantoin (%)	2.6±0.01	1.0±0.02
Na (mg/g)	63.1±1.20	42.4±1.0
Mg (mg/g)	14.7±0.48	6.9±0.18
K (mg/g)	13.1±0.28	8.1±0.10
Ca (mg/g)	26.40±0.38	66.7±1.46
Cu (mg/kg)	364.3±21.10	265.9±40.12
Zn (mg/kg)	42.6±2.90	34.0±5.12
Fe (mg/kg)	66.3±8.68	30.6±4.47
Mn (mg/kg)	21.4±1.22	2.8±0.44
Cl ⁻ (mg/g)	93.2±1.09	66.6±0.86
SO ₄ ²⁻ (mg/g)	14.8±0.65	16.4±0.36
NO ₃ ⁻ (mg/g)	0.16±0.01	1.1±0.08
Glycolic acid (mg/g)	7.1±0.41	1.0±0.18

4) Do the authors have standardized the snail mucus collection? Are all the batches comparable in the present compounds? This is an important point to be sure of the standardized efficacy of the final product. This is important also in order to make the research to be reproducible.

Our Response and Revision: Thanks for the reviewer's kindly suggestion. The snail mucus from all batches were prepared with the same standardized procedure, as shown below. This procedure was also described in the Method section. We also compared the main components (protein, s-GAG, and Na) of d-SMG from random 3 batches, which showed very small discrepancy (Supplementary Table 9).

Additionally, according to the suggestion of reviewer 2, we also supplemented the assays to test the effect of s-GAG on wound healing in normal rats, alongside with that of d-SMG. For d-SMG, these new results were consistent with the previous results shown in our old manuscript, possibly indicating the stable and reproducible efficacy of d-SMG.

Standard preparation procedure of d-SMG

Supplementary Table 9. Component content of three batches of d-SMGs.

Component		Protein (%)	Basic amino acid (% in protein)	Aromatic amino acid (% in protein)	s-GAG (%)	Na (%)
Species & Batches						
	Batch-1	34.17	17.13	10.24	15.68	6.49
A. fulica d-SMG	Batch-2	32.71	16.57	10.40	15.90	6.27
	Batch-3	32.64	17.25	10.52	16.15	6.17
	Batch-1	37.81	15.14	10.67	9.37	4.39
H. lucorum d-SMG	Batch-2	43.11	15.45	10.63	9.38	4.11
	Batch-3	37.75	15.83	11.11	9.01	4.19

Taking into consideration the above-mentioned concerns, the manuscript needs a careful revision, taking into consideration the additional information needed.

In summary, we tried our best to improve the manuscript and made some changes marked in red in the revised manuscript. These changes will not influence the framework of the paper. We appreciate for Editors/Reviewers' warm work earnestly, and hope that the correction will meet with approval.

REVIEWER COMMENTS

Reviewer #1 (Remarks to the Author):

The authors have made significant improvements to the manuscript. They have enriched the text with key discussion points, highlighted limitations, and added many additional strong datasets that support the physical and biological mechanism of action of the material. The manuscript has been greatly improved and is publishable after some of my remaining concerns are addressed.

It is wonderful to see the mass spec data and the protein composition of the SMG. I notice that there is no mention of mucins in this gel. Was that surprising to the authors? Is it because there are no mucins in these gels because the snail equivalent is far genetically from other mammals, or perhaps because they are difficult to detect?

Question 8.

The following reference: Yao, M., Wei, Z., Li, J. et al. Microgel reinforced zwitterionic hydrogel coating for blood-contacting biomedical devices. Nat. Commun. 13, 5339 (2022) does not seem to explain why 5% is an acceptable limit. Perhaps it is not an absolute limit you should look for, but a relative performance relative to a known biocompatible compound?

Question 9.

Could the authors explain what the n=10 refers to? Are those technical repeats, or biological repeats? Could the authors indicate the 100% labelled on the y-axis of the MTT graph? Since you have measured the growth of the cell numbers over time, it would be cleaner to normalise the metabolic activity by the cell number of each condition at each timepoint instead of normalising to control.

Question 10.

The authors claim the concentration of d-SMG is 0.2 mg/mL. Typically, a range of concentrations is tested to find the toxic concentration. Could the authors explain why/how 0.2 mg/mL is relevant here?

Question 11.

Thank you for clarifying what the GAG controls material is. This is helpful to understand the work. With this knowledge, could the authors explain then how the property of SMG differs from the extracted s-GAG?

Question 13.

It is not clear to me if the decrease in the number of vessels from day 7 to 14 was expected. Why would this happen? This might require some discussion. (Fig 6I)

Question 14.

I understand that the authors claim the good performance of Alginate Figure 7 could be due to the ability of alginate to remove liquid and trap cytokines. But how would that compare to the SMG, which is likely hydrated and can, as the authors have well demonstrated, can also bind strongly to cytokines.

If anything, this looks like it could be an artifact in the experiments and should be disclaimed.

Reviewer #2 (Remarks to the Author):

Accepted

Reviewer #3 (Remarks to the Author):

The authors have revised the manuscript according to the reviewer's requests.

REVIEWER COMMENTS

Our Response: The editors and reviewers' work are greatly appreciated. The following are point-by-point responses to the reviewers' comments, and according to their suggestions, we have revised the manuscript, with changes marked red in the resubmitted text. We hope that after the revision the manuscript is acceptable.

The authors have made significant improvements to the manuscript. They have enriched the text with key discussion points, highlighted limitations, and added many additional strong datasets that support the physical and biological mechanism of action of the material. The manuscript has been greatly improved and is publishable after some of my remaining concerns are addressed.

Our Response: We thank the reviewer for his/her constructive comments and suggestions, which are important to improve the manuscript.

It is wonderful to see the mass spec data and the protein composition of the SMG. I notice that there is no mention of mucins in this gel. Was that surprising to the authors? Is it because there are no mucins in these gels because the snail equivalent is far genetically from other mammals, or perhaps because they are difficult to detect?

Our Response: Thanks for the reviewer's comments on the protein composition of SMG. Indeed, mucins are a common component in most gel-like secretions, but no mucins were identified in our previous analysis. Previously, the peptides degraded from SMG were identified by searching the database (<https://www.uniprot.org/uniprotkb?query=Stylommatophora>), which contains 88348 proteins in total, including only 19 mucins, from the Order Stylommatophora (land snail). Based on this database no mucin was identified from our mass spec data.

Nevertheless, the review's comment also roused our curiosity, therefore, we reanalyzed our mass spectrometry data using a more specialized database (<https://www.uniprot.org/uniprotkb?query=Gastropoda%20mucin>), which includes 142 mucins from the Class Gastropoda. By searching this specialized database, 6 mucins and 10 mucins were identified from *A. fulica* d-SMG and *H. lucorum* d-SMG, respectively. We have added these data in the revised manuscript (Supplementary Table 5–6).

Supplementary Table 5. Identified mucins in the *A. fulica* d-SMG by LC-MS/MS.

Protein IDs (Uniprot)	Protein	Organism	Unique peptides	Unique sequence coverage	Protein abundance
A0A2C9JYP9	Mucin-2-like	Biomphalaria glabrata	1	3.4%	70.67%
A0A433TRR6	Mucin 96d	Elysia chlorotica	1	16.1%	12.40%
A0A2T7PKE9	Mucin-2-like	Pomacea canaliculata	4	14.8%	9.15%
A0A2T7PBD6	Mucin-2-like	Pomacea canaliculata	1	2.8%	5.69%
A0A2C9L226	Mucin-like protein	Biomphalaria glabrata	1	3.7%	1.25%
A0A3S0ZR63	Conserved secreted mucin	Elysia chlorotica	2	3.7%	0.85%

Supplementary Table 6. Identified mucins in the *H. lucorum* d-SMG by LC-MS/MS.

Protein IDs (Uniprot)	Protein	Organism	Unique peptides	Unique sequence coverage	Protein abundance
A0A2T7NL50	Mucin-2-like	Pomacea canaliculata	1	0.6%	70.12%
A0A0B6ZJM7	Mucin-5AC	Arion vulgaris	2	1.3%	7.98%
A0A2T7NM56	Mucin-5AC-like	Pomacea canaliculata	1	3.0%	7.51%
A0A0B6ZSG9	Mucin-5AC-like	Arion vulgaris	1	0.9%	3.70%
A0A2C9LWW2	Mucin-1	Biomphalaria glabrata	1	3.1%	2.75%
A0A2T7PRF9	Mucin-like protein	Pomacea canaliculata	2	1.9%	1.87%
V4AEI7	Mucin 5B	Lottia gigantea	1	13.2%	1.74%
A0A2T7PR58	Mucin-like protein	Pomacea canaliculata	1	0.7%	1.72%
A0A2C9KEQ3	Mucin-13	Biomphalaria glabrata	1	4.2%	1.41%
A0A2T7PHX2	Mucin-5AC-like	Pomacea canaliculata	2	2.8%	1.21%

Question 8.

The following reference: Yao, M., Wei, Z., Li, J. et al. Microgel reinforced zwitterionic hydrogel coating for blood-contacting biomedical devices. Nat. Commun. 13, 5339 (2022) does not seem to explain why 5% is an acceptable limit. Perhaps it is not an absolute limit you should look for, but a relative performance relative to a known biocompatible compound?

Our Response and Revision: Thanks for the reviewer's careful checking. We have removed the reference, and revised the description. According to Standard Practice for Assessment of Hemolytic Properties of Materials (ASTM F 756-17, 2017) issued by American Society for Testing and Materials (ASTM), the hemocompatibility of materials are classified into three levels: haemolytic (haemolysis over 5%), slightly haemolytic (between 5% and 2%), non-haemolytic (below 2%). Additionally, the Chinese Pharmacopeia (2020) set the standard of non-hemolysis by qualitative analysis compared with normal saline. Based on these standards, no obvious hemolysis effect of d-SMG was observed or detected (<5%, calculated by the equation:

$$\text{Hemolysis ratio (\%)} = \frac{A_{\text{sample}} - A_{\text{control}}}{A_{\text{positive}} - A_{\text{control}}} \times 100 \%$$

where A_{sample} , A_{control} , and A_{positive} represent the absorbance of the sample, negative control, and positive control, respectively).

Question 9.

Could the authors explain what the n=10 refers to? Are those technical repeats, or biological repeats? Could the authors indicate the 100% labelled on the y-axis of the MTT graph? Since you have measured the growth of the cell numbers over time, it would be cleaner to normalise the metabolic activity by the cell number of each condition at each timepoint instead of normalising to control.

Our Response and Revision: Thanks for the reviewer's patient review and comments to improve our manuscript. In previous assay, for each group the same batch of cells were seeded into 10 micro-wells ($n = 10$), so it should be

technical repeats. Considering both Question 9 and 10, we decided to further improve this assay, including more biological repeats and different concentrations of d-SMG.

Briefly, three batches of frozen L929 cells were recovered and subcultured for 3 generations. Then each batch of cells was seeded in a 96-well microplate, after cultivation for 24 h, the complete medium was replaced by the new medium with or without d-SMG (0.05, 0.1, 0.2, 0.5, 1.0, or 2.0 mg/mL), and after cells were cultured for another 24 h, 48 h or 72 h, the cell viability was measured by MTT method. Each treatment was conducted in 3 repeated wells (technical repeats) in the evaluation for each batch of cells. The above processes were independently evaluated with three batches of cells (biological repeats, $n = 3$). The results are shown in Figure 3a. We have revised the related Method and Result in the new manuscript.

Fig. 3a. Effect of different concentrations (0.05, 0.1, 0.2, 0.5, 1.0, and 2.0 mg/mL from left to right in the graph) of d-SMG on cell viability by MTT method. Data were presented as the mean \pm SEM, $n = 3$, biological repeats.

For the cell growth assay (Supplementary Fig. 9a), cells in each well were continuously monitored and counted. As the reviewer suggested, we have revised Supplementary Fig. 9a.

Supplementary Fig. 9a. Effect of d-SMGs on the cell growth of L929. Data were normalized to the cell count before treatment (0 h), and expressed as mean \pm SEM, $n = 5$ (technical repeats).

Question 10.

The authors claim the concentration of d-SMG is 0.2 mg/mL. Typically, a range of concentrations is tested to find the toxic concentration. Could the authors explain why/how 0.2 mg/mL is relevant here?

Our Response and Revision: Thanks for the reviewer's comment. We studied the effect of a series concentrations (0.05–2.0 mg/ml) of d-SMG on cell viability. The results showed that d-SMG from both *A. fulica* and *H. lucorum* at the concentration of 0.05–1.0 mg/ml had no effect on the cell viability. And for the group treated with d-SMG up to 2.0 mg/ml (in complete medium), the cell viability was higher than 80%. When d-SMG with concentration higher than 2.0 mg/ml it became sticky gel, which is difficult to be quantitatively applied to cell assay. We have supplemented the data in the revised manuscript (Fig. 3a).

Question 11.

Thank you for clarifying what the GAG controls material is. This is helpful to understand the work. With this knowledge, could the authors explain then how the property of SMG differs from the extracted s-GAG?

Our Response: SMG is a natural biological gel material, mainly composed of protein and polysaccharide. And s-GAG is the glycosaminoglycan purified from SMG. Based on their composition and physicochemical property, SMG as a bioadhesive can promote wound healing by sealing wound, absorbing exudate, and regulating the inflammatory environment in chronic wound. Whereas, the polysaccharide s-GAG, as the main active component of SMG, is responsible for the inflammation modulating function of SMG (s-GAG itself does not have potent adhesion). We have added these discussions in the revised manuscript.

Question 13.

It is not clear to me if the decrease in the number of vessels from day 7 to 14 was expected. Why would this happen? This might require some discussion. (Fig 6I)

Our Response: Thanks for the review's insightful comment. According to the pathophysiological process of angiogenesis in wound healing (as shown below), in the proliferative phase hypoxia stimulates the production of pro-angiogenic factors (VEGF, FGF, ANG2), resulting in the sprouting of immature and disorganized new capillaries. Then in the remodeling phase, anti-angiogenic factors (pigment epithelium derived factor (PEDF) and Sprouty-2) can cause most of the newly formed capillaries to undergo apoptosis, thus the neocapillary bed is pruned back to normal density with attendant maturation. Maturation factors (ANG1, PDGF, α -SMA) support the recruitment of stabilizing pericytes and the maturation of the basement membrane on the new capillaries. The result is a stable, well perfused capillary bed with a vessel density similar to normal uninjured tissue. (Han, C. *et al.* Angiogenesis in wound repair: too much of a good thing? *Cold Spring Harbor Perspect. Biol.* 14, a041225 (2022); Okonkwo, U. A. *et al.* Diabetes and wound angiogenesis. *Int. J. Mol. Sci.* 18, 1419 (2017)). In **Fig 6i**, the wound healing in day 7 and 14 may be in the proliferative and remodeling phase, respectively, that's why the vessel number in day 14 seemed decreased when compared with that in day 7. We have added this discussion in the revised manuscript.

Reference Figure: Time course and pattern of wound angiogenesis (Han, C. et al. A. Angiogenesis in wound repair: too much of a good thing? Cold Spring Harbor Perspect. Biol. 14, a041225 (2022)).

Question 14.

I understand that the authors claim the good performance of Alginate Figure 7 could be due to the ability of alginate to remove liquid and trap cytokines. But how would that compare to the SMG, which is likely hydrated and can, as the authors have well demonstrated, can also bind strongly to cytokines. If anything, this looks like it could be an artifact in the experiments and should be disclaimed.

Our Response: It's thought-provoking comment. Indeed, d-SMG can be hydrated, and can also absorb liquid or exudate. But unlike alginate dressing, due its biodegradability d-SMG was not removed from the wound because of its biodegradability when collecting the wound tissue samples for cytokine analysis (Fig. 7d). Therefore, the reduction of inflammatory cytokines in d-SMG treated group should be due to its activity in reducing cytokine secretion, which is associated with its direct or indirect (amelioration of the inflammatory microenvironment due to cytokine capture) effect on promoting macrophage polarization to the M2 anti-inflammatory phenotype. Additionally, during the experiment d-SMG was applied to the wound as a single dose, while alginate was replaced every 2–3 days according to its product instruction. Additionally, Alginate derived from seaweed was a commercial biomaterial as a common-used wound dressing, but it requires complex processing and has poor biodegradability. The natural biomaterial d-SMG, which is readily obtained from snail, exhibits excellent biodegradability and promotes wound healing.

Reviewer #2 (Remarks to the Author):

Accepted

Our Response: We are thankful for the reviewer's time and effort to improve the manuscript.

Reviewer #3 (Remarks to the Author):

The authors have revised the manuscript according to the reviewer's requests.

Our Response: We are thankful for the reviewer's positive comments on our work and important contributions to improve the manuscript.

REVIEWERS' COMMENTS

Reviewer #1 (Remarks to the Author):

The answers provided make good sense. I have no further comments. Congratulations to the authors for their great work.

REVIEWERS' COMMENTS

Reviewer #1 (Remarks to the Author):

The answers provided make good sense. I have no further comments. Congratulations to the authors for their great work.

Our Response: We are thankful to the reviewer for his/her positive comments on our work and for his/her important contributions to improve the manuscript.